# Mammalian antiviral proteins ZAP and KHNYN can independently restrict CpG-enriched avian viruses

Jordan T. Becker[1‡*], Clayton K. Mickelson[1], Lauren M. Pross[1], Autumn E. Sanders[1], Esther R. Vogt[1], Frances K. Shepherd[1], Chloe Wick[2], Alison J. Barkhymer[1], Stephanie L. Aron[1], Elizabeth J. Fay[1,2], Reuben S. Harris[3,4*], Ryan A. Langlois[1*]

**1** Department of Microbiology and Immunology, University of Minnesota—Twin Cities, Minneapolis, Minnesota, United States of America, **2** Department of Biochemistry, Molecular Biology, and Biophysics, University of Minnesota—Twin Cities, Minneapolis, Minnesota, United States of America, **3** Department of Biochemistry and Structural Biology, University of Texas Health, San Antonio, Texas, United States of America, **4** Howard Hughes Medical Institute, University of Texas Health, San Antonio, Texas, United States of America

‡ Lead contact.
* beck1169@umn.edu (JTB); langlois@umn.edu (RSH); rsh@uthscsa.edu (RAL)

## Abstract

Zoonotic viruses are an omnipresent threat to global health. Influenza A virus (IAV) transmits between birds, livestock, and humans. Proviral host factors involved in the cross-species interface are well known. Less is known about antiviral mechanisms that suppress IAV zoonoses. We observed CpG dinucleotide depletion in human IAV relative to avian IAV. Notably, human ZAP selectively depletes CpG-enriched viral RNAs with its cofactor KHNYN. ZAP is conserved in tetrapods, but we uncovered that avian species lack KHNYN. We found that chicken ZAP may not affect IAV (PR8) or CpG-enriched IAV (PR8$_{CG}$). Human ZAP or KHNYN independently restricted CpG-enriched IAV PR8$_{CG}$ by overexpression in chicken cells and by combined knockout in human cells. Additionally, mammalian ZAP-L and KHNYN also independently restricted an avian retrovirus (ROSV). Curiously, platypus KHNYN, the most divergent from eutherian mammals, was also capable of independent restriction of multiple diverse viruses. We suggest that some mammalian KHNYN can act as a *bona fide* restriction factor with cell-autonomous activity. Furthermore, we speculate that through repeated contact between avian viruses and mammalian hosts, protein changes may accompany CpG-biased mutations or reassortment to evade mammalian ZAP and KHNYN.

## Introduction

Influenza A virus (IAV) is a constantly circulating infectious burden in global waterfowl populations, agricultural animals, and among humans seasonally. Avian IAV

**Data availability statement:** All relevant data are within the paper, its Supporting information files, and/or a data repository (https://doi.org/10.5281/zenodo.14919910). RNA-seq data are available from the NCBI Sequence Read Archive (RNA-seq data; accession number PRJNA1208514; https://ncbi.nlm.nih.gov.bioproject/PRJNA1208514).

**Funding:** This study was funded by the National Institute of Allergy and Infectious Disease (NIAID - https://www.niaid.nih.gov) grants R56-AI150965 (to R.A.L.), R01-AI148669 (to R.A.L.), and F32-AI147813 (to J.T.B.) as well as NIAID Centers for Excellence in Influenza Research and Response (CEIRR - https://www.ceirr-network.org) grant NHH75N93021C00017 (to R.A.L.). R.S.H. is an Investigator of the Howard Hughes Medical Institute and the Ewing Halsell President's Council Distinguished Chair. The funders played no role in study design, data collection and analysis, decision to publish, or preparation of the manuscript.

**Competing interests:** The authors have declared that no competing interests exist.

**Abbreviations:** BSD, blasticidin S deaminase; CDS, coding sequences; dN/dS, nonsynonymous/synonymous codon; FBS, fetal bovine serum; ggaZAP, chicken ZAP; H3, histone 3; HIV-1, human immunodeficiency virus 1; hpi, post-infection; hZF2, human ZF2; IAV, influenza A virus; IBV, influenza B virus; ICV, influenza C virus; IFNα, interferon-alpha; IP, immunoprecipitated; iPARP, inactive poly-ADP ribose polymerase; ISGs, interferon stimulated genes; kb, kilobases; MLV, murine leukemia virus; mNG, mNeonGreen-transduced; mNG-ggaZAP, mNeonGreen-tagged chicken ZAP; MOI, multiplicity of infection; nt, nucleotides; PCA, principal component analysis; RABV-G, rabies virus glycoprotein; RMSD, root mean squared deviation; ROSV, Rous sarcoma virus; sgRNA, single guide RNA; SRA, Sequence Read Archive; TIDE, tracking of indels by decomposition; TUBA, tubulin; vRNA, viral RNA; YFP, yellow fluorescent protein; ZAP, zinc finger antiviral protein; ZAP-L, long isoform of human ZAP; ZAP-S, short isoform of human ZAP; ZF2, zinc finger 2.

occasionally transmits into domesticated mammals and humans. While these infections do not usually result in successful human-to-human spread, they are distinctly pathogenic with a reportedly high mortality rate [1]. However, when novel avian IAV establishes in the human population there can be devastating consequences. Major pandemics have been the result of avian-to-human zoonoses or mixed-animal zoonoses involving avian-swine-human infection chains [2]. Notably, multiple pandemics or large-scale epidemics occurred in 1918, 1957, 1968, and 2009 that resulted in massive infection incidence and mortality in human populations [3]. Much is known about the host dependency factors to which avian IAV must adapt to infect human cells productively [4]. These include differences in viral hemagglutinin binding to host sialic acid linkages [5] with species-specific airway localization [6], pH-sensitivity of viral RNA (vRNA) endosomal release into the cytoplasm [7], and species-specific ability to use the polymerase cofactor, ANP32 [8]. Relatively less is known about the network of restriction factors that might specifically target zoonotic IAV infections in mammals [4]. Much of our knowledge of host-pathogen interactions and genetic conflicts has focused on amino acid level and related protein structure changes that predicate animal-to-human adaptation [9]. Comparatively less is known about nucleic acid adaptations driven by host-pathogen conflicts during interspecies infections.

Viral nucleic acid sequence and structure are not only a determinant of protein coding changes, but rather they encompass a complex and sensitive network of features that determine viral fitness, stability, and pathogenesis [10,11]. In addition, host and pathogen genomes are not in equilibrium regarding the frequency and usage of nucleotide doublets [12,13]. For example, CpG dinucleotide frequency is depleted throughout the human genome [14]. Commonly, this is a result of DNA methylation at CpG motifs near promoter sequences that regulate transcription [15]. Mammalian genomes have also selected against encoding CpG dinucleotides in cytoplasmic-resident RNAs [14, 16]. Others observed a depletion of CpG content among interferon stimulated genes (ISGs), notably in mammalian species [14,17]. Accordingly, many viral pathogens that infect mammalian species match the dinucleotide content of their host species and mammalian viruses are often CpG-depleted [12,14,18,19]. Others showed that avian IAV sequences exhibit elevated CpG content and that CpG content has depleted over time for IAV strains circulating in humans [14,19]. This suggests that host factors might exist in human hosts to select against CpG-enriched viral nucleic acids that are absent or defective in avian species.

Recently, several groups showed that the human restriction factor, zinc-finger antiviral protein (ZAP, also known as PARP13), specifically binds to and suppresses CpG dinucleotides in cytoplasmic RNA [20,21]. Human ZAP is encoded by the *ZC3HAV1* gene, induced by interferon, and generates multiple splice isoforms [22,23], including ZAP-L (a 902 amino acid isoform preferentially localized to intracellular membranes) and ZAP-S (a 699 amino acid isoform with diffuse cytoplasmic localization). Like many antiviral proteins, ZAP is evolving under positive selection [24–26]. ZAP antiviral activity is regulated by the interferon-inducible E3 ubiquitin ligase, TRIM25 [20,21,27,28]. KHNYN was identified as a ZAP-interacting protein and putative cofactor for ZAP antiviral activity [21]. Moreover, KHNYN exhibits basal expression,

localizes throughout the cytoplasm [29], and is not interferon-inducible [30]. While KHNYN can exert antiviral effects through ZAP, it is unknown if it can exert additional direct antiviral functions. Together, human ZAP, KHNYN, and TRIM25 may function cooperatively, selectively, and/or independently to restrict certain classes of viruses with a bias towards viruses with CpG-enriched nucleotide content [31]. While the function of these human proteins has been well studied, less is known about their nonprimate and nonmammalian orthologues and their antiviral restriction ability.

Understanding the intrinsic and innate immune factors capable of restricting zoonotic viruses is paramount in identifying zoonotic viruses with pandemic potential [32]. Importantly, IAV is a seasonal infectious burden in humans and common zoonosis among wild waterfowl, agricultural birds, domesticated mammals, and humans [4]. We hypothesized that birds might be defective in RNA recognition and restriction pathways and this defect presents a barrier to emergence in mammals. Here, we tested whether restriction factors targeting CpG-enriched RNAs could affect replication of IAV with an elevated CpG content, mimicking avian IAV. We found that chicken and duck ZAP do not restrict the common laboratory IAV strain (A/Puerto Rico/8/1934/H1N1; PR8) replication regardless of CpG content. We show through genomic, phylogenetic, and functional analyses that multiple avian species may be largely defective in CpG-targeting ZAP function. Additionally, we demonstrate that avian species lack KHNYN, which likely arose through gene duplication after birds and mammals diverged [33,34]. Thus, we used chicken cells as a gain-of-restriction system to test the exogenous provision of host factors for viral restriction. Indeed, human ZAP-S and KHNYN stably expressed in chicken cells independently restricted CpG-enriched PR8 IAV replication. We showed that combined knockout of ZAP and KHNYN in human cells increased replication of CpG-enriched PR8 IAV to a greater extent than single knockout of ZAP or KHNYN alone. We also found that mammalian but not avian (*Galloanserae*) ZAP, as well as some mammalian KHNYN proteins can restrict a naturally CpG-enriched avian retrovirus, Rous sarcoma virus (ROSV). Unexpectedly, we discovered a potently antiviral KHNYN homologue from platypus that suggests an ancient restriction pathway. We show that KHNYN may be a *bona fide* restriction factor with cell-autonomous activity. Furthermore, we speculate that through repeated contact between avian and human hosts followed by seasonal circulation among humans, protein level changes for receptor preference and polymerase function could be accompanied by dinucleotide changes and/or CpG-disfavored reassortment that evade restriction by mammalian ZAP and/or KHNYN.

## Results

### Mammalian IAV sequences are CpG-depleted relative to avian IAV

Viruses evolve to replicate successfully in their host species. Previous work showed that human IAV exhibited a depletion of CpG dinucleotide content relative to avian IAV sequences since the 1918 IAV pandemic [14,19,35]. Here, surveying influenza sequences in FluDB [36] collected between 1918 and 2020, we also observed a depletion of CpG dinucleotides per kilobase (CpG content) in human and swine but not avian IAV sequences over time since 1918 (S1A Fig). By comparing all human, swine, and avian IAV segment sequences *en masse*, we observed a lower CpG content in human and swine IAV relative to avian IAV sequences (Fig 1A). We observed similar results for $\rho_{CG}$ (rho [$\rho$] accounts for total C and G content [13,14,17]) but less so for other dinucleotides or %GC content (S2A–S2D Fig). Notably, the largely human-specific influenza B virus [37] (IBV) and influenza C virus [38] (ICV) are CpG-depleted relative to all IAV sequences, speculatively consistent with long-term near-exclusive circulation in mammalian/human populations (Figs 1A and S1–S2). We observed that human IAV segments 1–3 encoding the heterotrimeric viral polymerase (i.e., PB2, PB1, and PA, respectively) are more CpG-depleted relative to avian IAV than segments 4–6 encoding virion proteins (i.e., HA, NP, and NA, respectively), typically subject to adaptive immune pressure [39] (S1B Fig). We found that when comparing major HA and NA subtypes, CpG content is depleted in subtypes seasonally circulating in humans (H1N1 and H3N2) relative to avian sequences of the same subtype (S1C Fig). Conversely, CpG content in human isolates of common avian-to-human zoonotic HA and NA subtypes (e.g., H5N1 and H7N9) is similar to their avian counterparts. The natural history of human IAV includes multiple

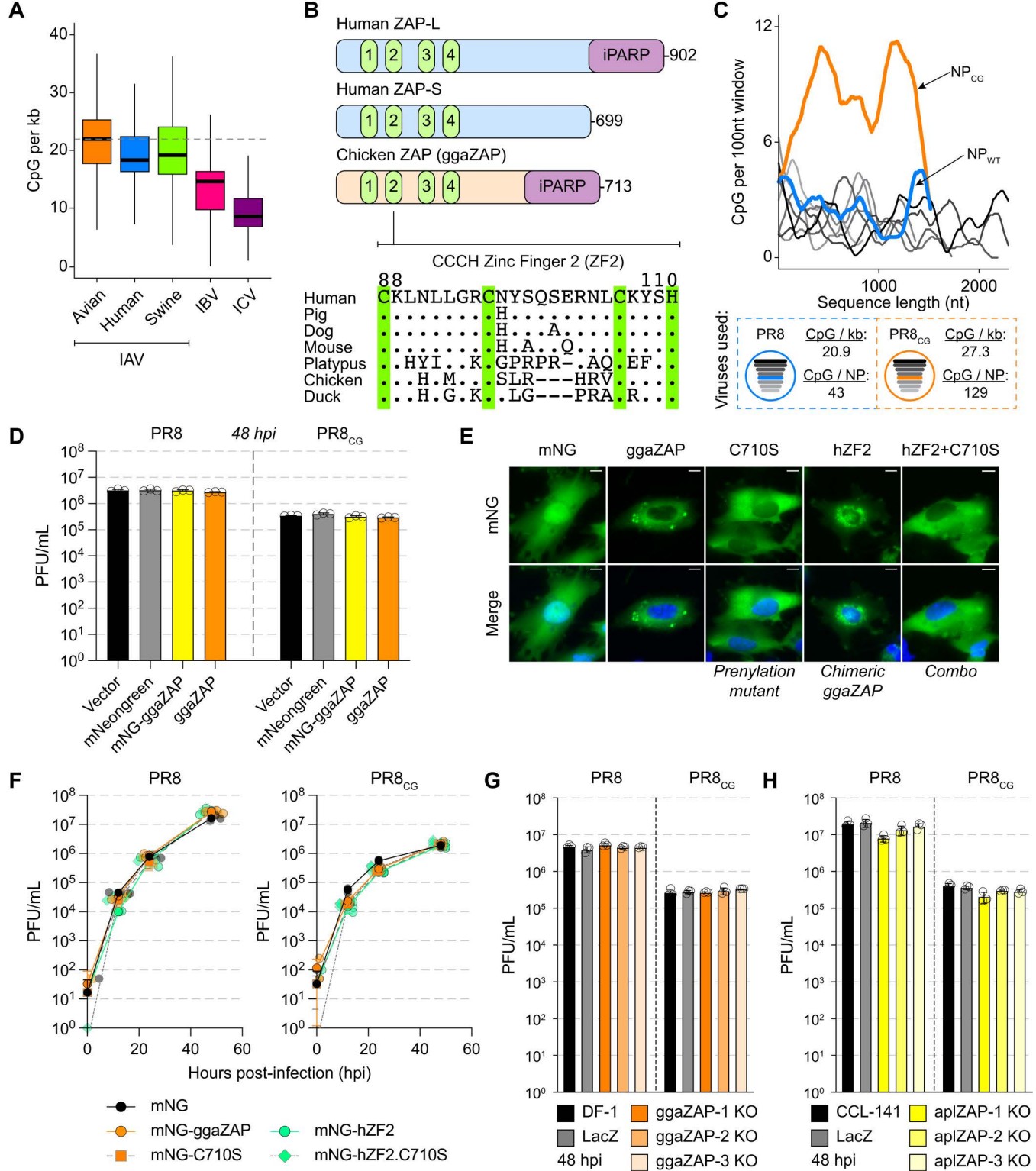

**Fig 1. Chicken and duck ZAP do not potently restrict replication of PR8 IAV or CpG-enriched PR8 IAV. (A)** IAV sequences exhibit depletion of CpG content (CpG per kb = # CpG/segment length in kilobases [kb]). Box and whisker plots showing CpG content of all IAV sequences from avian

(orange), human (blue), and swine (green) hosts as well as IBV (pink) and ICV (purple). Thick black bars represent median CpG content, boxes define the 25–75th percentile, and whiskers represent range excluding outliers. Horizontal dashed gray lines highlight median avian IAV CpG content as a reference. The data underlying this panel can be found in https://doi.org/10.5281/zenodo.14919910. **(B)** Schematic of human ZAP-L, human ZAP-S, and chicken ZAP (ggaZAP) protein domain architecture including four CCCH zinc fingers and inactive poly-ADP ribose polymerase (iPARP). Amino acid alignment of ZAP zinc finger 2 (ZF2; human residues 88–110) from multiple species. Conserved CCCH residues are highlighted with green vertical bars. **(C)** CpG content of PR8 (shades of gray by segment length), wild-type NP (blue), and CpG-enriched NP (orange) calculated as # CpG per 100 nucleotides (nt) over a sliding window (top). Schematic of PR8 and PR8$_{CG}$ viruses and displaying CpG content for each virus and total CpG number encoded by each NP segment (bottom). **(D)** Titers of PR8 and PR8$_{CG}$ at 48 hours post-infection (hpi) in chicken DF-1 cells stably expressing mNeon-Green (mNG, gray), mNG-tagged chicken ZAP (mNG-ggaZAP, yellow), or untagged ggaZAP (orange) as well as control transduced cells (Vector, black). Bar represents mean plaque-forming units per mL (PFU/mL) with error bars representing standard deviation of the mean. Multiplicity of infection (MOI) = 0.05 (pfu per cell). Individual data points are displayed as black circle outlines. **(E)** Fluorescence images of DF-1 cells expressing mNG as well as fusion proteins of mNG-ggaZAP and mutants indicated (green, top). Nuclei counterstained with DAPI (blue) and shown in merge image (bottom). Scale bars represent 10 microns. **(F)** Titers of PR8 and PR8$_{CG}$ over 48 hpi in chicken DF-1 cells stably expressing mNG (black circles), mNG-ggaZAP (orange circles), mNG-hZF2 (green circles), mNG-C710S (orange/gray squares), and mNG-hZF2.C710S (green/gray diamonds) mutants shown in (E). MOI = 0.05. Individual data points are displayed as 50% transparent jittered shapes behind mean data point with errors bars representing standard deviation of the mean. **(G)** Titers of PR8 and PR8$_{CG}$ at 48 hpi in parental untransduced DF-1 cells (DF-1, black), pooled CRISPR/Cas9 knockout chicken DF-1 cells expressing a control sgRNA targeting *Escherichia coli* LacZ (LacZ, gray), or sgRNAs targeting endogenous chicken ZC3HAV1 (ggaZAP1-3 KO, shades of orange). Bar represents mean PFU/mL with error bars representing standard deviation of the mean. MOI = 0.05 (pfu per cell). Individual data points are displayed as black circle outlines. **(H)** Titers of PR8 and PR8$_{CG}$ at 48 hpi in parental untransduced CCL-141 cells (CCL-141, black), pooled CRISPR/Cas9 knockout duck CCL-141 cells expressing a control sgRNA targeting *E. coli* LacZ (LacZ, gray), or sgRNAs targeting endogenous duck ZC3HAV1 (aplZAP1-3 KO, shades of yellow). Bar represents mean PFU/mL with error bars representing standard deviation of the mean. MOI = 0.05 (pfu per cell). Individual data points are displayed as black circle outlines. The data underlying the graphs in this figure can be found in S2 Table.

introductions [40] resulting from avian zoonoses and/or reassortment of avian IAV segments with human viruses in 1918, 1957, 1968, and 2009. We compared CpG content of avian IAV to human IAV during the time periods surrounding the replacement of H1N1 with H2N2 in 1957, H2N2 with H3N2 in 1968, reintroduction of H1N1 in 1977, and pandemic "swine flu" H1N1 in 2009 (S1D Fig, bottom). We observed that CpG content is lower in human relative to avian IAV sequences during the longer time periods following pandemics in 1977 and 2009 (S1D Fig, top). This may reflect periodic reintroduction of CpG-enriched avian sequences that deplete over longer time frames.

## Chicken and duck ZAP may not restrict replication of CpG-enriched IAV

We hypothesized that avian species are defective in cytoplasmic CpG-sensing by ZAP and reasoned that this deficiency allows CpG-maintenance of IAV sequences in avian host species. Human ZAP-L and chicken ZAP are homologous proteins encoding four zinc fingers, an inactive poly-ADP-ribose polymerase-like domain [41], and carboxy-terminal pre-nylation motif [23,29] (Fig 1B, top). Previous structural and functional analysis of human ZAP identified zinc finger 2 (ZF2, residues 88–110) as responsible for directly recognizing and binding to CpG dinucleotides in the context of short (3–20 nucleotide) and long (full-length 9 kilobase HIV-1 genome) RNA molecules [20,42,43]. Using two structural prediction algorithms, we generated models of the four zinc fingers of chicken ZAP (ggaZAP, residues 1–220) by both comparative modeling based on human ZAP (RobettaCM [44], PDB: 6UEJ [42]) and ab initio folding by RoseTTAFold [45] (S3A and S3B Fig). We also compared these structures to the model of chicken ZAP predicted by AlphaFold2 [46] (S3A and S3B Fig). As expected, we observed that both ab initio predicted structures (RoseTTAFold and AlphaFold2) exhibited lower structural similarity (as measured by root mean squared deviation, RMSD) to human ZAP relative to those generated with comparative modeling (S3B Fig). In addition, we observed structural differences between human ZAP and models of duck ZAP as well as quail ZAP (S3A and S3B Fig). Alignment of the amino acid sequences of ZAP homologs revealed multiple amino acid differences between ZF2 of avian and mammalian species (Fig 1B, bottom). Therefore, we hypothesized that chicken ZAP is unable to restrict CpG-enriched viruses and tested this hypothesis using the common IAV laboratory strain A/Puerto Rico/8/1934//H1N1 (PR8, blue dashed box) and a CpG-enriched version of PR8 [47] (PR8$_{CG}$, orange dashed box; Fig 1C).

In chicken DF-1 cells stably expressing chicken ZAP (ggaZAP) as well as mNeonGreen-tagged chicken ZAP (mNG-ggaZAP), we found no difference in virus replication over 48 hours for PR8 or PR8$_{CG}$ relative to vector-transduced (Vector) and mNeonGreen-transduced (mNG) cells (Fig 1D). PR8$_{CG}$ exhibits consistently lower replication relative to wild-type PR8 as previously shown [47], but allows comparisons for viral replication across different cellular contexts. We confirmed expression of transduced proteins by western blot for mNG as well as blasticidin S deaminase (BSD) [48,49] expressed on the bicistronic mRNA and used for drug selection (S3C Fig). The chicken *ZC3HAV1* gene encodes a single ZAP isoform of 713 amino acids with a carboxy-terminal prenylation motif (amino acids: $_{710}CIVC_{713}$) that is homologous to the long isoform of human ZAP (ZAP-L). The short isoform of human ZAP (ZAP-S) is generated by alternative splicing and lacks the carboxy-terminal prenylation motif and the inactive poly-ADP-ribose polymerase-like domain. To test if either prenylation of chicken ZAP or if the divergent ZF2 regulate restriction of CpG-enriched PR8 IAV, we generated mutants of chicken ZAP lacking the critical cysteine residue (C710S) for membrane-targeting prenylation, a chimera encoding the human ZF2 (hZF2), or a combination (hZF2.C710S) of these two mutants all of which expressed and localized as expected (Figs 1E and S3D). However, no form of chicken ZAP restricted PR8 or PR8$_{CG}$ replication (Fig 1F). We confirmed protein expression of transduced genes by western blot for mNG as well as BSD (S3D Fig). Finally, we found that pooled CRISPR/Cas9-mediated knockout of endogenous ZAP in chicken DF-1 and duck CCL-141 cells had minimal effects on PR8 or PR8$_{CG}$ replication (Figs 1G–1H and S4). Similarly, we observed that knockdown of endogenous ZAP by stable shRNA in chicken DF-1 cells or transient siRNA in duck CCL-141 cells had minimal effects on PR8 or PR8$_{CG}$ replication (S3E–S3G Fig). qPCR showed endogenous chicken ZAP is expressed in DF-1 cells, induced by interferon, and can be suppressed by shRNA (S3E Fig). Attempts at co-expression of human ZAP-S and KHNYN as exogenous transgenes in chicken DF-1 cells were complicated by apparent ZAP-S-related inhibition of KHNYN protein expression or stability (S3H Fig). Given that human ZAP can use chicken TRIM25 as a cofactor [50], we reasoned that chicken DF-1 cells are potentially deficient in ZAP-dependent antiviral activity targeting this CpG-enriched IAV and could serve as a gain-of-restriction system to test related host factors for the ability to restrict CpG-enriched viruses.

## Human ZAP-S and KHNYN can independently restrict CpG-enriched PR8 IAV

Restriction of CpG RNA has been attributed to human ZAP along with its cofactors KHNYN and/or TRIM25. However, we found that avian species may lack a *bona fide* KHNYN orthologue. There are no annotated avian KHNYN orthologues in ensembl, NCBI, or UCSC Genome Browser and a ~400,000 bp syntenic region surrounding human KHNYN is missing in avian species, including the KHNYN paralogue—NYNRIN (S5A Fig). Human KHNYN and NYNRIN are encoded in tandem on chromosome 14 and NYNRIN is the result of a KHNYN gene duplication and fusion to a retroviral integrase domain [34] (S5B Fig). KHNYN and NYNRIN are paralogous to N4BP1 [51], a gene conserved in birds, reptiles, and fish (S5B and S5C Fig). We estimated a maximum likelihood phylogeny of human KHNYN, NYNRIN, and N4BP1 orthologues nucleotide sequences from ensembl (S5C Fig). We confirmed that N4BP1 is conserved in *Gnathostomats* (i.e., jawed vertebrates including fish, birds, reptiles, and mammals) and is closely related to a fish-specific N4BP1 paralogue (sometimes annotated as *khnyn* or *khnyn-like*, e.g., *Danio rerio khnyn*, ENSDARG00000092488). However, KHNYN found in mammalian species (i.e., monotremes, marsupials, and placentals) and NYNRIN found in therian mammals (i.e., marsupials and placentals) branched together and are more like each other than to the N4BP1 genes (S5C Fig). We noted that NCBI and ensembl contain reptile sequences annotated as *KHNYN* (e.g., *Alligator mississippiensis KHNYN* XM_014607035.3 or *Anolis carolinensis khnyn* ENSACAG00000009105) that may represent ancestral KHNYN orthologues based on nucleotide sequence (S5C Fig; ancient KHNYN orthologues), exon organization, and syntenic neighborhood (S5D Fig). Thus, we reasoned that avian species may be defective and/or deficient in two restriction factor genes: ZAP as well as KHNYN. As such, we generated chicken DF-1 cells that stably expressed human ZAP-L, ZAP-S, KHNYN, or TRIM25. We included a mutant of KHNYN that is expected to be enzymatically inactive within its endonuclease domain [21] due to substitution of three critical aspartic acid residues (DDD443/524/525AAA, dKHNYN). We found that ZAP-S as well as KHNYN modestly restricted PR8 relative to vector transduced cells (Fig 2A, left). KHNYN has not been shown previously

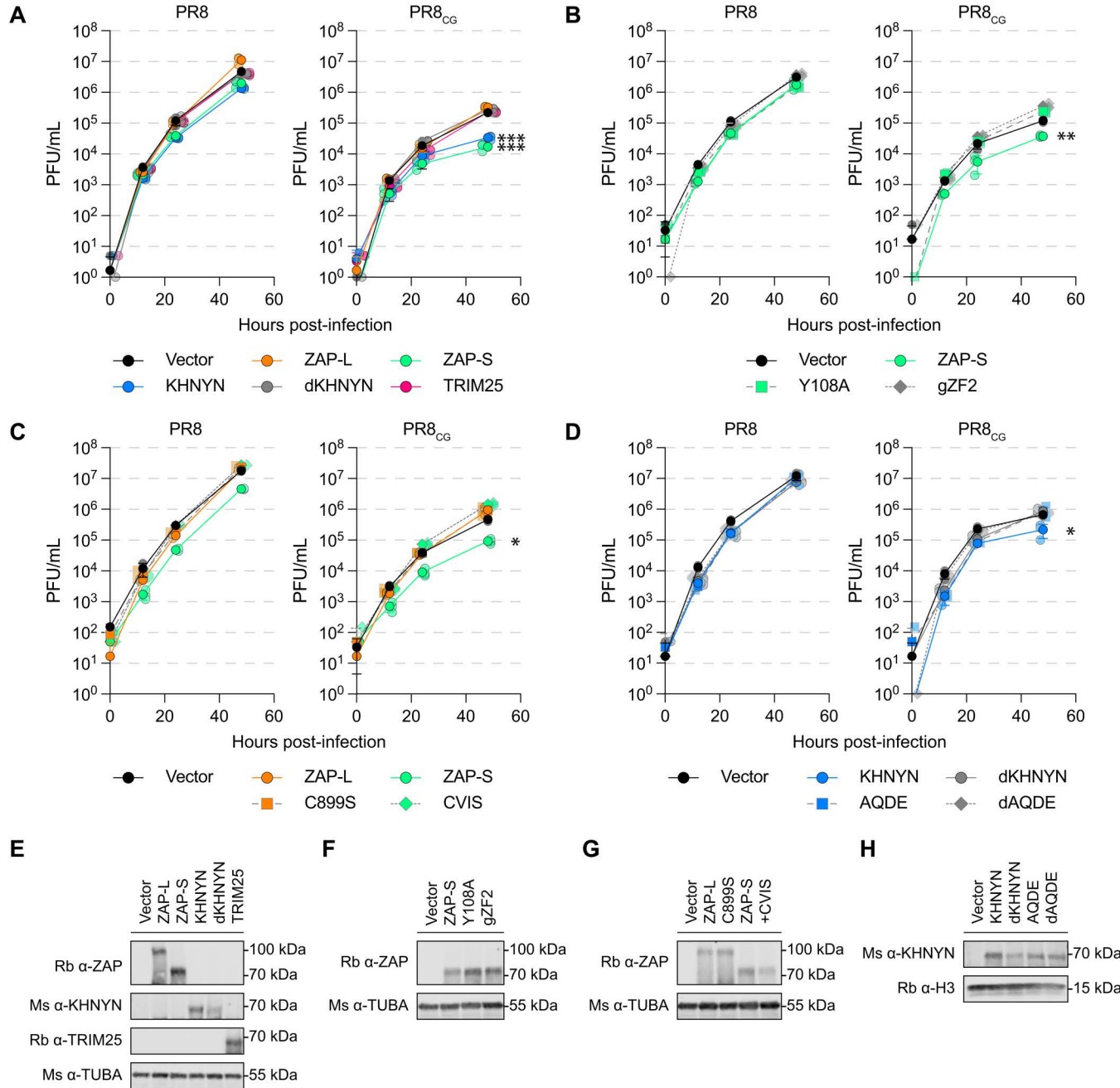

**Fig 2. Human ZAP-S and KHNYN can independently restrict CpG-enriched PR8 IAV. (A)** Titers of PR8 and PR8$_{CG}$ over 48 hpi in chicken DF-1 cells stably expressing human ZAP-L (orange circles), ZAP-S (green circles), KHNYN (blue circles), dKHNYN (gray circles), or TRIM25 (pink circles) relative to Vector (black circles). MOI = 0.05. *** indicates $p \leq 0.0001$ at 48 hpi by two-way ANOVA. **(B)** Titers of PR8 and PR8$_{CG}$ over 48 hpi in chicken DF-1 cells stably expressing RNA-binding mutants of ZAP-S (green circles) including Y108A (green squares, dashed line), and chimeric encoding chicken ZF2 (gZF2; gray diamonds, dashed line) relative to Vector (black circles). MOI = 0.05. ** indicates $p = 0.0093$ at 48 hpi by two-way ANOVA. **(C)** Titers of PR8 and PR8$_{CG}$ over 48 hpi in chicken DF-1 cells stably expressing C-terminal mutants of human ZAP-L (orange circles) and ZAP-S (green circles) including ZAP-L C899S (oranges squares, dashed line) and ZAP-S +CVIS motif (CVIS; green diamonds, dashed line). MOI = 0.05. * indicates $p = 0.0108$ at 48 hpi by two-way ANOVA. **(D)** Titers of PR8 and PR8$_{CG}$ over 48 hpi in chicken DF-1 cells stably expressing mutants of KHNYN (blue circles) including catalytically inactive dKHNYN (gray circles), $_{95}$AQ/DE$_{96}$ mutant (AQDE; blue squares, dashed line), and combination mutant of dKHNYN with AQDE (dAQDE; gray diamond, dashed line). MOI = 0.05. * indicates $p = 0.0158$ at 48 hpi by two-way ANOVA. **(E–H)** Western blots of chicken DF-1 cells stably expressing human ZAP, KHNYN, or TRIM25 and mutants used in (A–D) detecting ZAP, KHNYN, and/or TRIM25 as well as tubulin (TUBA) or histone 3 (H3) loading

controls. Individual data points are displayed as 50% transparent jittered shapes behind mean data point with errors bars representing standard deviation of the mean. The data underlying the graphs in this figure can be found in S2 Table. Original annotated immunoblot images can be found in S1 Raw Images.

to act independently of ZAP. Surprisingly, we found that human ZAP-S and KHNYN (but not ZAP-L, dKHNYN, or TRIM25) independently restricted PR8$_{CG}$ relative to vector (Fig 2A, right).

## CpG-targeted restriction requires intact RNA-binding domain of human ZAP and extended di-KH domain of KHNYN

To confirm the role of ZAP-S and KHNYN in restricting PR8$_{CG}$, we generated mutants of each, stably expressed them in DF-1 cells, and performed multicycle IAV replication assays. To test the CpG specificity of ZAP-S in restricting PR8$_{CG}$, we generated a Y108A mutant previously shown to reduce CpG specificity [42] as well as a chimeric human ZAP-S encoding the ZF2 of chicken ZAP (gZF2). Only WT human ZAP-S remained capable of restricting PR8$_{CG}$, an ability that was lost by the Y108A and gZF2 mutants (Fig 2B). To test if the prenylation motif (amino acids: CVIS) and corresponding membrane localization of ZAP regulated restriction activity against PR8$_{CG}$, we generated a mutant of ZAP-S that encoded a carboxy-terminal CVIS motif (+CVIS) as well as a C899S mutant of ZAP-L [23]. While the C899S mutant of human ZAP-L did not gain the ability to restrict PR8$_{CG}$, human ZAP-S +CVIS lost the ability to restrict PR8$_{CG}$ as expected (Fig 2C). Next, to delineate the manner of KHNYN restriction of PR8$_{CG}$, we mutated its extended di-KH domain [52]. KH-domain containing proteins encode a GXXG amino acid motif at the top of a central alpha-helix and mutations in the GXXG motif can reduce RNA-interactions in another KH-domain containing protein, HNRNPK [53]. We generated models of this extended di-KH domain (amino acids 1−213) of KHNYN by ab initio folding (RoseTTAFold) (S5E Fig) that showed a central alpha-helix with the $_{94}$GAQG$_{97}$ motif at its apex adjacent to a three-helix bundle (S5E Fig). This model appeared structurally similar to a recent crystal structure of KHNYN extended di-KH domain [52]. We found that only WT human KHNYN was capable of restricting PR8$_{CG}$ over 48 h of replication in chicken DF-1 cells, but not the catalytically inactive dKHNYN, a AQ95/96DE mutant (AQDE), or a catalytically inactive AQ95/96DE (dAQDE) (Fig 2D). We confirmed expression of transduced human proteins by western blot (Fig 2E–2H). The Vector transduced cells used in Fig 1D showing chicken ZAP had minimal, if any, effects on PR8 or PR8$_{CG}$ are the same Vector transduced cells used throughout Fig 2A–2D showing human ZAP-S and KHNYN restricted PR8$_{CG}$ allowing for relative comparisons in ability for these proteins to restrict these IAV. Nonetheless, we wanted to compare human ZAP-S restriction of PR8 or PR8$_{CG}$ to chicken ZAP directly in multicycle IAV replication assays. Using chicken DF-1 cells stably expressing yellow fluorescent protein (YFP) tagged proteins, we found that human ZAP-S and KHNYN restricted PR8$_{CG}$ while human ZAP-L, chicken ZAP, and dKHNYN failed to negatively affect PR8 or PR8$_{CG}$ (S6A and S6B Fig). Human ZAP-S and KHNYN also modestly restricted an avian IAV strain (OH175) with a modest but naturally CpG-enriched PA segment (S6A–S6C Fig). Finally, we observed that human ZAP-S, KHNYN, and dKHNYN may independently interact with PR8$_{CG}$ segment 5 RNA by RNA immunoprecipitation and quantitative PCR (RIP-qPCR) in infected DF-1 cells stably expressing these YFP-tagged proteins (S6D Fig). We also observed similar expression of viral proteins (NP) in PR8 and PR8$_{CG}$ infected cells. Together, these results indicated that intact human ZAP-S as well as KHNYN can independently restrict PR8$_{CG}$.

## Chicken and IAV mRNA are largely unaffected by human ZAP or KHNYN

To understand how these mammalian restriction factors might affect IAV, we also examined transcriptomic changes during IAV infection. We performed RNA-seq on uninfected as well as PR8 or PR8$_{CG}$ infected chicken DF-1 cells stably expressing human ZAP-L, ZAP-S, KHNYN, or dKHNYN to evaluate if these human proteins modulated the chicken transcriptome and viral mRNA expression relative to vector transduced cells. We found that the transcriptomes of uninfected cells were

largely unperturbed by whatever human factor was overexpressed (S7A Fig) suggesting antiviral activity may not be caused by changing the underlying gene expression profile. Next, we hypothesized that by interacting with and depleting CpG-enriched viral RNAs during multicycle IAV replication experiments, ZAP-S and KHNYN might also negatively bias CpG-enriched segment accumulation during replication. We evaluated relative levels of IAV mRNA segments in PR8 versus PR8$_{CG}$ infected cells in the presence of these different human factors and found similar levels of viral mRNA within segments (S7B Fig). Infection with either PR8 or PR8$_{CG}$ resulted in similar gene expression patterns for host cell factors between the two viruses, suggesting that the mechanism of restriction is not through differential induction of host antiviral responses (S7C Fig). Together, these results suggested initial mechanistic insights into how ZAP and KHNYN might or might not restrict IAV during zoonotic infections in that the restriction may not substantially deplete viral mRNAs.

## Endogenous human ZAP and KHNYN restrict CpG-enriched viruses

We next tested the ability of endogenous human ZAP and/or KHNYN to restrict CpG-enriched IAV in human lung cells. Therefore, we generated multiple human lung adenocarcinoma A549 clonal cell lines depleted for ZAP, KHNYN, or both by CRISPR/Cas9 genome editing as well as a vector control (Vector) and a negative control guide targeting *Escherichia coli* LacZ beta-galactosidase (LacZ) clonal cell lines (Fig 3A–3B). We tested PR8 and PR8$_{CG}$ by multicycle replication assay in these cell lines. We found that single knockout of only ZAP or only KHNYN failed to increase PR8 or PR8$_{CG}$ replication relative to LacZ or parental A549 cells (Fig 3C). We reasoned that this was consistent with our results by over-expression in DF-1 cells such that either ZAP or KHNYN are capable of independently restricting PR8$_{CG}$. Therefore, we tested multicycle replication of PR8 and PR8$_{CG}$ in double knockout (ZAP and KHNYN depleted, ZK1–2) cells revealing that genetic loss of both ZAP and KHNYN increased replication of PR8$_{CG}$ relative to parental A549 or Vector cell lines (Fig 3D) supporting the hypothesis that ZAP and KHNYN can independently restrict of CpG-enriched IAV. We tested two single-cell clones per gene targeted for knockout to assess potential clonal effects and we hypothesize that the decreased replication in ZAP KO 2 is a clonal effect. The same guide (ZAP guide 2) was used to make the double knockouts that allowed for increased replication across two independent clones. We also found replication of PR8 or PR8$_{CG}$ was unaffected by knockout of endogenous human TRIM25 in A549 cells (Fig 3E). Altogether, these results support the hypothesis that ZAP and KHNYN may act independently against CpG-enriched PR8 IAV.

## ROSV is CpG-enriched and susceptible to mammalian CpG-dependent restriction

Avian IAV exhibits an enrichment of CpG content relative to human IAV (Figs 1A, S1, and S2). We next asked if other avian viruses with different levels of human or mammalian zoonotic interactions might exhibit higher levels of CpG enrichment. We noticed that ROSV (a tractable retrovirus model in chicken) exhibits a higher CpG content relative to mammalian retroviruses, including human immunodeficiency virus 1 (HIV-1) or murine leukemia virus (MLV). We observed that ROSV is CpG-enriched along its entire length relative to mammalian retroviruses including HIV-1 or MLV (Fig 4A). To determine whether ROSV was subject to CpG-dependent restriction, we performed retroviral assembly and single-cycle infectivity assays (schematic in Fig 4B) by co-transfecting chicken DF-1 cells with a full-length ROSV reporter virus expression plasmid and a VSV-G expression plasmid along with increasing amounts of control, human ZAP-L, human ZAP-S, KHNYN, or dKHNYN plasmid. We found that human ZAP-L inhibited ROSV infectivity, but ZAP-S did not (Fig 4C). Using mutants of human ZAP-L and ZAP-S, we confirmed that for ROSV, ZAP restriction required an intact ZF2 and prenylation motif (Fig 4C). Specifically, the C899S prenylation mutant of ZAP-L failed to restrict ROSV while ZAP-S fused to the four amino acid ZAP-L prenylation motif (+CVIS) gained the ability to restrict ROSV. Next, we tested human ZAP-L, chicken ZAP, and duck ZAP as well as prenylation mutants of each for restriction of ROSV using fusions to mNeongreen. We found that human ZAP-L restricted ROSV but neither of the avian ZAP proteins or their prenylation mutants were much less restrictive (Fig 4D). Finally, we tested mammalian orthologues of ZAP from dog, pig, and platypus and found that the prenylated forms of placental mammal ZAP (pig and dog) restricted ROSV, while platypus ZAP or a prenylation

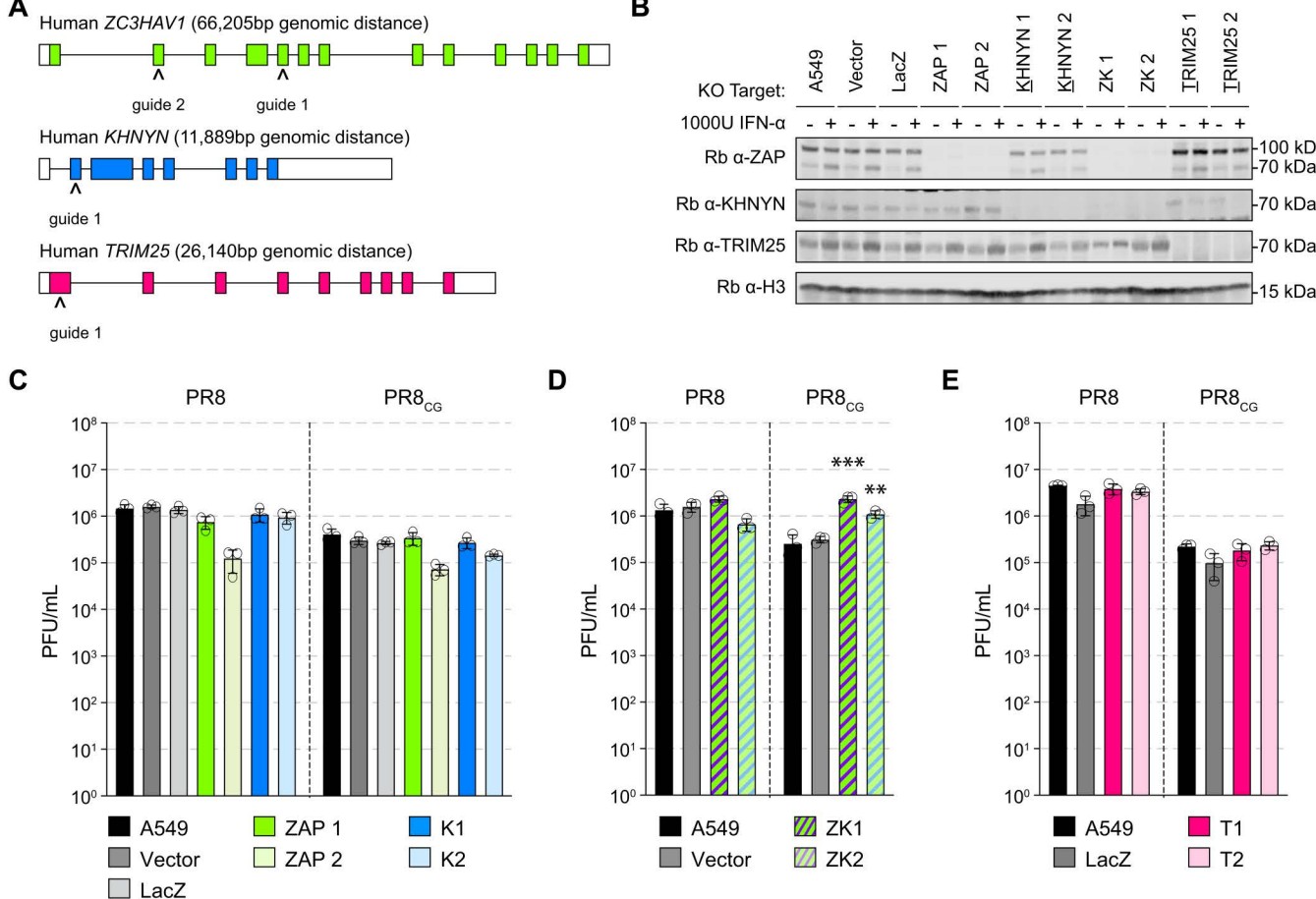

**Fig 3. Endogenous human ZAP and KHNYN can independently restrict CpG-enriched PR8 IAV. (A)** Schematic depicting exon organization of human ZAP (green), KHNYN (blue), and TRIM25 (pink) genes and sites targeted by CRISPR single guide RNAs (sgRNAs; carets). Colored boxes indicate coding exons and white boxes indicate noncoding portions of exons. **(B)** Western blots of CRISPR/Cas9 engineered knockout (KO) human A549 cell lines including parental A549 cells, control clone transduced with pLentiCRISPR1000 (Vector), control clone targeting *Escherichia coli* LacZ (LacZ), two ZAP KO clones (ZAP1 and ZAP2), two KHNYN KO clones (K1 and K2), two ZAP and KHNYN dual KO clones (ZK1 and ZK2), and two TRIM25 KO clones (T1 and T2). Cells were treated with vehicle (PBS) or 1,000 units per mL (U/mL) universal interferon-alpha (IFNα) for 24 hours. Histone H3 detected as loading control. **(C)** Titers of PR8 and PR8$_{CG}$ at 48 hpi in human A549 KO cells including parental (black), Vector and LacZ (shades of gray), ZAP KO clones (shades of green), and KHNYN KO clones (shades of blue). MOI = 0.05. **(D)** Titers of PR8 and PR8$_{CG}$ at 48 hpi in human A549 KO cells including parental (black), Vector (gray), and ZAP+KHNYN KO clones (green and blue striped). MOI = 0.05. *** indicates $p \le 0.0001$ and ** indicates $p = 0.0045$ by one-way ANOVA. **(E)** Titers of PR8 and PR8$_{CG}$ at 48 hpi in human A549 KO cells including parental (black), LacZ (gray), and TRIM25 KO clones (shades of pink). MOI = 0.05. Bars represent mean PFU/mL with error bars representing standard deviation of the mean. Individual data points are displayed as black circle outlines. The data underlying the graphs in this figure can be found in S2 Table. Original annotated immunoblot images can be found in S1 Raw Images.

mutant (C653S) were less restrictive though they expressed less readily (S8A Fig). We confirmed expected subcellular localization of mammalian and avian ZAP isoforms and mutants in chicken DF-1 cells (S9A Fig) and observed conservation of CaaX prenylation amino acid motif of representative ZAP orthologues (S9B Fig).

## Mammalian KHNYN restricts ROSV independent of ZAP

Next, we tested human KHNYN, mutant KHNYN, and mammalian KHNYN orthologues from pig, dog, and platypus for restriction of ROSV. Pigs and dogs are domesticated mammals frequently in contact with agricultural poultry, a common

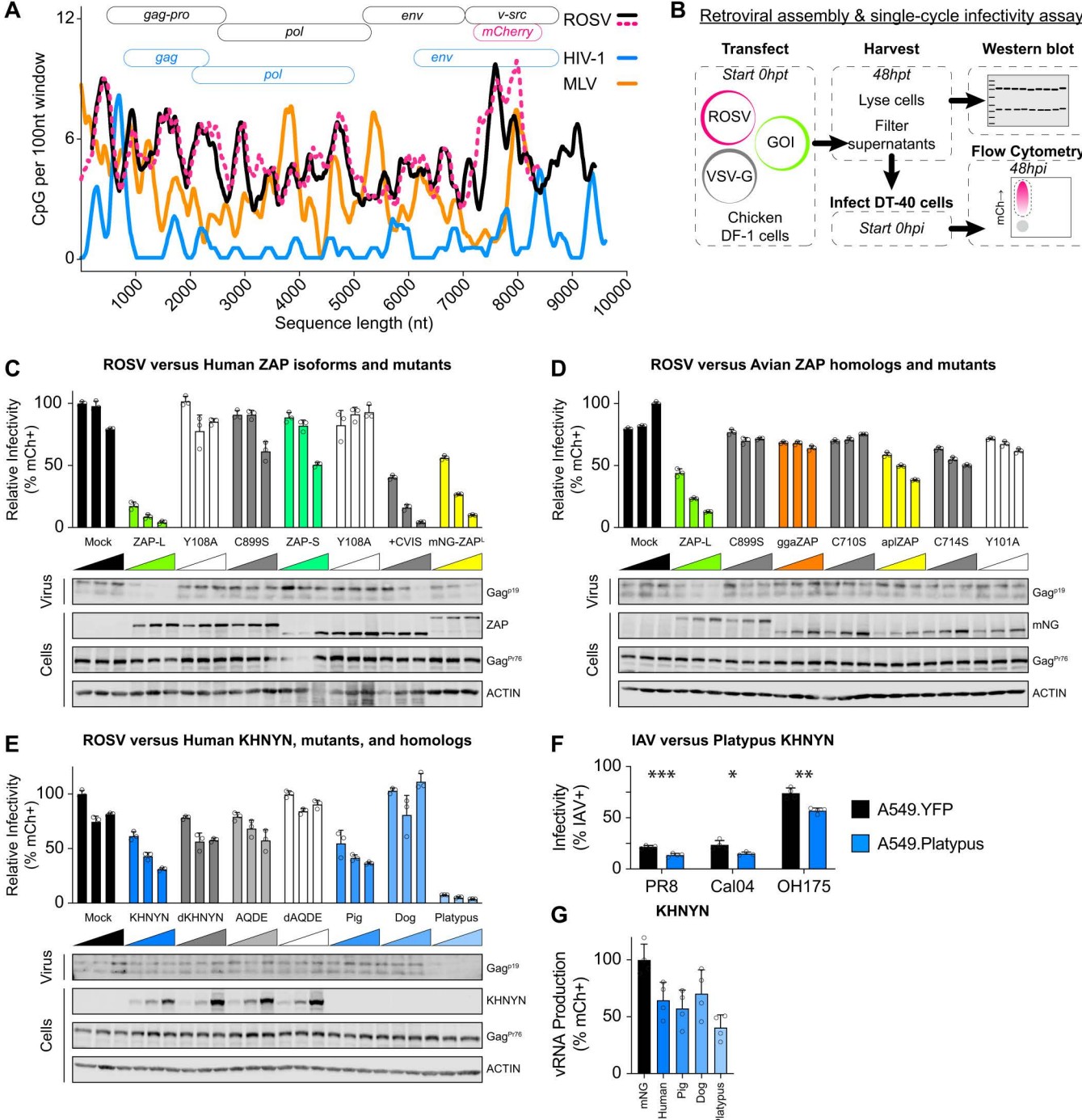

**Fig 4. Prenylated human ZAP and mammalian KHNYN can independently restrict naturally CpG-enriched ROSV. (A)** CpG content of retroviruses (ROSV in black, ROSV-mCherry reporter virus in pink dashed line, MLV in orange, and HIV-1 in blue) calculated as # CpG per 100 nucleotides over a sliding window (graph). Schematic of retrovirus open reading frame (ORF) organization (top). **(B)** Schematic of retroviral assembly and single-cycle infectivity assay including plasmid transfection into cells, cell lysate and supernatant harvest at 48 hours post-transfection (hpt), storage, and western blotting as well as DT40 suspension cell infection and flow cytometry as a measure of infectivity. **(C)** Retroviral assembly and single-cycle infectivity results of human ZAP-L, ZAP-S, and mutants effects on ROSV showing relative percent ROSV infected cells (% mCherry-positive; % mCh+) as determined by flow cytometry. DF-1 cells transfected with increasing plasmid amounts of indicated host proteins (100, 200, or 400 ng), 1,500 ng ROSV, and 200 ng VSV-G. Infectivity is relative to backbone pcDNA3.2 transfections (Mock). Below, western blots detecting ROSV cleaved Gag$^{p19}$ in supernatant

virions as well as full-length ROSV Gag^Pr76, ZAP proteins, and ACTIN loading controls from cellular lysates. **(D)** Retroviral assembly and single-cycle infectivity results of human ZAP-L, chicken (gga) ZAP, duck (apl) ZAP, and prenylation mutant on ROSV infectivity. Below, western blots detecting ROSV cleaved Gag^p19 in supernatant virions as well as full-length ROSV Gag^Pr76, mNG-tagged ZAP proteins, and ACTIN loading controls from cellular lysates. **(E)** Retroviral assembly and single-cycle infectivity results of human KHNYN, mutants, and mammalian KHNYN homologs (from pig, dog, and platypus) on ROSV infectivity. Below, western blots detecting ROSV cleaved Gag^p19 in supernatant virions as well as full-length ROSV Gag^Pr76, human KHNYN proteins, and ACTIN loading controls from cellular lysates. Note mammalian KHNYN orthologues were not detected by human KHNYN antibodies. **(F)** Single-cycle infectivity (%mCh+ or %NP+) in A549 stable cells expressing YFP-alone (black) and YFP-tagged platypus KHNYN (blue) effects on single-cycle IAV including laboratory PR8, pandemic Cal04, and avian-origin OH175. *** indicates $p = 0.000832$, * indicates $p = 0.025875$, and ** indicates $p = 0.002263$ by unpaired $t$ test. **(G)** IAV Minigenome reporter assay comparing mNG-tagged mammalian KHNYN (shades of blue: human, pig, dog, and platypus) relative to mNG alone (black) effects on IAV vRNA production (% mCh+ cells) using PB2, PB1, PA, and NP expression plasmids and a negative-sense mCherry vRNA reporter plasmid (encoding 34 CpG dinucleotides within the proxy vRNA). The bar represents mean relative infectivity with error bars representing standard deviation of the mean. Individual data points are displayed as black circle outlines. The data underlying the graphs in this figure can be found in S2 Table. Original annotated immunoblot images can be found in S1 Raw Images.

source of avian-to-mammal zoonoses [4]. We selected platypus KHNYN to test because it is the only mammalian species to encode KHNYN but lack the NYNRIN gene [34] (S3C Fig). We found that pig KHNYN, but not dog KHNYN, similarly restricted ROSV as human KHNYN while platypus KHNYN potently restricted ROSV (Fig 4E). We also tested these proteins (with mNeongreen-tagged at their amino terminus) relative to catalytically inactivated mutants for ROSV restriction activity (S8B Fig). Curiously, mutation of the aspartic acid residues in the NYN-endonuclease domains inhibited the restrictive ability for pig KHNYN (DD508/508AA) but not platypus KHNYN (DDD497/498AAA) (S8B Fig). We tested multiple truncated forms of platypus KHNYN and found that truncation mutants which fully deleted the endonuclease domain lost restriction activity (S8C Fig). We observed that the NYN domains of platypus KHNYN, including the aspartic acid catalytic residues, are relatively well conserved by amino acid alignment and computational structural modeling (S10A and S10B Fig). We generated structural models of platypus KHNYN and noted a substantial structural difference in the loop above the apex of the putative GXXG motif in the KH domain in platypus KHNYN encoding a notably shorter loop relative to human, pig, and dog KHNYN (S5E Fig). We confirmed appropriate subcellular localization of these KHNYN orthologues in chicken DF-1 cells (S10C Fig). Curiously, we found that platypus KHNYN also restricted HIV-1, a CpG-enriched HIV-1, and MLV, independent of glycoprotein provided for pseudo-typing (S8D Fig). We detected cellular expression of retroviral (ROSV Gag^Pr76 and Gag^p19; HIV-1 Gag^Pr55 and Gag^p24; MLV Gag^mCh) and antiviral proteins for retroviral assembly and single-cycle infectivity assays as well as loading controls and purified virions from culture supernatants. We noted that platypus KHNYN could potently inhibit virion assembly and release of ROSV, HIV-1, and MLV with typically minimal effects on intracellular Gag expression or loading controls (notably in Figs 4E and S8D). We also found similar results for mammalian KHNYN restriction of ROSV in the absence of chicken ZAP using DF-1 cells with *ggaZC3HAV1* knocked out by CRISPR/Cas9 (S8E and S4 Figs). Finally, we tested if platypus KHNYN could restrict IAV replication in human A549 cell lines expressing YFP-tagged platypus KHNYN. Relative to YFP-alone expressing cells, platypus KHNYN modestly inhibited single-cycle PR8, pandemic Cal04, and avian-origin OH175 (Fig 4F). Platypus KHNYN also restricted PR8_CG and to a lesser extent OH175 in multicycle replication assays (S6A Fig). We also observed that mammalian KHNYN negatively affects viral polymerase activity using an IAV minigenome reporter assay (Fig 4G). Together, these results support the hypothesis that mammalian ZAP and KHNYN can restrict CpG-enriched avian viruses, including strains of IAV as well as ROSV. Furthermore, we discovered an orthologue of KHNYN in platypus that may potently restrict multiple viruses from distinct virus families independent of CpG content.

## Discussion

KHNYN was identified as a ZAP-interacting cofactor that required ZAP to exert a combined antiviral effect against a CpG-enriched HIV-1 [21]. Human ZAP functions independent of KHNYN against multiple viruses including HIV-1 [20,29,54], Ebolavirus [55,56], Zikavirus [57], and Sindbis virus [28]. Here, we provide the first evidence that KHNYN

may be capable of exerting an antiviral effect independent of ZAP. We note that, by definition, restriction factors act in a cell-autonomous fashion [58,59]. Recently, two groups showed that KHNYN binds to RNA through its NYN domain rather than its extended di-KH domain, at least for the substrates tested [52,60]. As such, KHNYN may be a novel *bona fide* restriction factor with incompletely characterized RNA preferences. Furthermore, multiple groups showed different, sometimes conflicting, results regarding ZAP restriction of IAV [61–64]. Here, we observed that human IAV sequences exhibit a marked decrease in CpG content relative to avian IAV sequences (Figs 1A and S1, S2). We showed that chicken ZAP and duck ZAP may not restrict IAV, in particular PR8 or PR8$_{CG}$ (Figs 1 and S3, S4) or ROSV (Figs 4 and S8). We reasoned that chicken cells (e.g., DF-1) could be used as a gain-of-restriction system to test restriction factors for their ability to inhibit IAV and other CpG-enriched viruses. In DF-1 cells stably expressing human ZAP-L, ZAP-S, and KHNYN, a catalytically inactive KHNYN, or TRIM25, we found that ZAP-S and KHNYN modestly restricted PR8 and significantly restricted PR8$_{CG}$ (Fig 2). Human ZAP-S and KHNYN also restricted an avian strain (OH175) (S6 Fig). We also confirmed that this restriction required an intact CpG-recognizing ZF2 of ZAP or a catalytically active KHNYN. We found that human ZAP and KHNYN do not appreciably manipulate the host transcriptome in chicken cells or substantially impact IAV mRNA levels by RNA-seq (S7 Fig). We supported our hypothesis that each of these restriction mechanisms can act independently in human cells wherein only combined knockout of both ZAP and KHNYN in human A549 cells increased replication of PR8$_{CG}$ (Fig 3). We also found that an avian retrovirus is subject to ZAP and KHNYN restriction, ROSV (Figs 4 and S8–S10). Finally, we discovered a potent antiviral KHNYN homologue in platypus capable of restricting multiple divergent retroviruses as well as multiple strains (PR8, PR8$_{CG}$, Cal04, and avian OH175) of IAV (Figs 4, S6, and S8). Altogether, we hypothesize that these mammalian restriction factors can act against CpG-enriched viruses including avian viruses and may select for CpG-depleted nucleotide mutations.

Multiple groups have analyzed the dinucleotide content of influenza sequences, other viruses, as well as host transcripts [12,14,17,19]. Here, we analyzed influenza virus segment sequences from different host species collected over the past century and found the CpG content is depleted over time in mammalian species and that this occurs in polymerase segments more so than HA or NP segments (Fig 1). We speculate that over a decades-to-centuries long timeframe, CpG-depletion is a relatively slow adaptative process compared to single amino acid substitutions in PB2, PA, NP, NA, or HA traditionally associated with host species adaptation [65–70]. We observed that H1N1 and H3N2 human IAV sequences exhibited a lower CpG content than subtype-matched avian sequences. However, H5N1 and H7N9 sequences were essentially identical in CpG content between human and avian samples likely reflecting the limited amount of time and replication within human patients. Yet, it is unclear how CpG content regulates viral RNA intrinsic stability or viral replication. Recent work showed that human ZAP targets viral RNA molecules encoding a minimum threshold number of CpG dinucleotides within a length of RNA that is neither too dense nor too sparse [71]. This likely reflects a requirement for some minimum number of ZAP molecules to bind RNA and if CpG dinucleotides are too close these sites occlude each other while if too sparse or too distributed this dilutes a necessary yet undefined minimum of ZAP clustering. Here, we showed that human ZAP-S can target an artificially CpG-enriched PR8 IAV and to a lesser extent PR8 IAV. Future work should examine how CpG number, density (number per length), segment-specificity, and influenza RNA class (viral messenger, complementary, or genomic RNA) are targeted by ZAP perhaps using deep mutational libraries of ZAP zinc fingers and/or target RNAs. Furthermore, we focused on CpG dinucleotides yet other dinucleotide, trinucleotide, or polynucleotide motifs under- or over-represented in host-virus relationships should be examined systematically [72–75]. In addition, much of our knowledge on host-pathogen interactions has focused on Red Queen conflicts highlighted by protein-protein interactions and nonsynonymous/synonymous codon (dN/dS) based metrics of positive selection [9,76–79]. Future work should explore other molecular interactions as surfaces for evolutionary conflict (e.g., RNA-protein and RNA-RNA) [80–82].

Human ZAP-S and ZAP-L can restrict different viruses and human transcripts [83], including those associated with the interferon response [17], based on their differential cellular localization as determined by the carboxy-terminal prenylation

motif [23,29,64]. KHNYN was identified by protein-protein interaction with ZAP in a yeast two-hybrid screen and function-ally shown to be a required cofactor for ZAP restriction of an artificially CpG-enriched HIV-1 construct [21]. Subsequent work suggested that ZAP-based restriction of different CpG-enriched HIV-1 constructs was not uniformly KHNYN-dependent [54]. Curiously, one group showed that ZAP-L requires the C-terminal prenylation motif to restrict CpG-enriched HIV-1 [29] while another found that a C-terminally HA-tagged ZAP-L (that effectively eliminates prenylation) restricted different CpG-enriched HIV-1 mutants [20,50]. It is unclear whether the differences in CpG number and density or other technical differences might explain these differences. Previous work demonstrated that human ZAP-L inhibited IAV (including PR8) by binding viral PB2 and PA proteins but is counteracted by PB1 [62]. However, others found that human ZAP-S inhibits IAV (including PR8) by reducing viral mRNA levels but is counteracted by NS1 [63]. These two results predated identification of human ZAP preference for CpG dinucleotides. Finally, recent work in human A549 cells showed that ZAP-S inhibits PR8 with a CpG-enriched segment 1 but not PR8 with a CpG-enriched segment 5 [64]. That virus was not affected by KHNYN KO or TRIM25 KO. In addition, they found that PR8$_{CG}$ (Segment 5) was attenuated in human A549 cells but was not rescued by ZAP KO alone or KHNYN KO alone. They did find that a homo-polymeric stretch of nucleotides in Segment 5 could be mutated to relieve the attenuation of PR8$_{CG}$ in human A549 cells [84]. How-ever, combined knockout of ZAP and KHNYN in human A549 cells was not tested. Here, we show that human ZAP-S and KHNYN can independently restrict IAV with a CpG-enriched segment 5 (PR8$_{CG}$; Figs 2 and 3) while ZAP-L and KHNYN can independently restrict ROSV (Fig 4). In addition, we found that chicken ZAP was largely ineffective against IAV (PR8, PR8$_{CG}$, or OH175) by overexpression as well as knockout and knockdown (Figs 1, S3, and S6). Knockout and knockdown of duck ZAP similarly failed to affect PR8 or PR8$_{CG}$ replication (Figs 1 and S3). Previous work showed chicken ZAP was less specific for CpG dinucleotides within the context of human-chicken chimeric ZAP proteins [50]. Future work determin-ing the preferred nucleotide preferences of other ZAP orthologues as well as the amino acid, structural, and domain (zinc finger 1–4) determinants of nucleotide preferences will be critical in understanding host-viral ZAP-RNA interactions and evolution. Chicken and duck encode primarily prenylated isoforms of ZAP [85], homologous to ZAP-L. We found that only prenylated isoforms of mammalian ZAP and no forms of chicken or duck ZAP restricted ROSV (Figs 4 and S8). Curiously, the original discovery of ZAP-dependent restriction of MLV identified and used a form of ZAP lacking the carboxy-terminal prenylation motif [86]. We confirmed that mammalian (human, pig, and dog) and avian (chicken and duck) ZAP proteins localized as expected with long or prenylation motif encoding isoforms at endomembranes and short or prenylation mutants exhibiting diffuse cytoplasmic localization (S9 Fig). We also note that platypus ZAP is not particularly restrictive against ROSV and has a more diverse CaaX motif, however, mutation of C653S appeared to affect its localization. Future work should explore mammalian ZAP restriction of IAV, in particular common host or mixing vessel species (e.g., pig, cow, or dog).

KHNYN has not been previously shown to independently restrict any viruses. Here, we discovered platypus KHNYN as a putative potent and broadly acting antiviral restriction factor. However, it is unclear how mammalian KHNYN functions independently to inhibit IAV or ROSV and how platypus KHNYN functions against such diverse viruses. We show prelimi-nary evidence for human KHNYN interaction with IAV PR8$_{CG}$ Segment 5 RNA, yet platypus KHNYN only weakly enriched for this RNA, albeit at lower protein expression levels. In addition, we observed that platypus KHNYN in particular could inhibit the assembly and release of retroviral virions with typically minimal effects on cellular retroviral Gag expression or loading control proteins. We previously showed that artificial relocalization of HIV-1 viral genomic RNA could inhibit virion assembly and release from cells [87,88]. Our results here might suggest platypus KHNYN may accomplish similar inhi-bition against many retroviruses. We also found that platypus KHNYN could inhibit PR8$_{CG}$ and OH175 in multi-cycle and/or single-cycle IAV assays. It's unclear how or if the antiviral mechanisms of platypus KHNYN against IAV versus retrovi-ruses are similar, related, or distinct. However, many antiviral proteins can exert restriction on viruses through multiple or disparate mechanisms. Notably, APOBEC3 proteins can exhibit both deaminase-dependent and deaminase-independent restriction against multiple viruses [89–92]. In addition, IFITM3 inhibits IAV during virus entry but can inhibit MLV during

egress and glycoprotein incorporation into progeny virions [93,94]. We might speculate that platypus KHNYN potency might compensate for platypus ZAP impotency observed in our assays against ROSV. Future work examining their relative antiviral activities and interactions will be informative for understanding the evolution of these two restriction factors. In particular, we showed that platypus KHNYN may exhibit a shorter loop above the extended di-KH domain and future work using chimera or mutants might inform greater molecular details and mechanistic implications of platypus KHNYN antiviral activity. Regardless, platypus KHNYN may function as a *bona fide* cell-autonomous restriction factor or potentially use cofactors or cellular features that are conserved across 300 million years of evolution between humans and chickens. Altogether, we suggest that antiviral activities of ZAP and KHNYN might be context dependent. For example, a subset of viruses might be restricted by the membrane-associated ZAP-L and others restricted by the cytosolic ZAP-S because the antiviral activity of many restriction factors is localization-dependent. In addition, ZAP might restrict some sequences, KHNYN might restrict other sequences, and they may act together against yet a third group of sequences. ZAP and KHNYN may also exhibit overlapping and independent restriction preferences as shown in (Figs 2 and 3). In fact, the results from Sharp, and colleagues suggest mutants of a specific virus strain (e.g., PR8 with CpG-enriched Segment 1 versus Segment 5) may exhibit differential susceptibility to ZAP [64,84]. In the case of segmented viruses, including IAV, this suggests an impact on genetic drift (mutation) as well as genetic shift (reassortment) over time.

In contrast to ZAP, relatively less is known about the function, preferences, and regulation of KHNYN. KHNYN is one member of family of paralogous genes encoding a putative RNA-binding domain fused to a NYN-endonuclease domain [51]. Our phylogenetic analysis (S5 Fig) support the hypothesis and previous reports that KHNYN may be derived from N4BP1 occurring prior to the radiation of mammalian species and NYNRIN is the result of a gene duplication and fusion/integration of KHNYN to Metaviral *Pol* gene occurring between the emergence of monotreme and therian mammals [34]. We showed here that mutation of key aspartic acid residues in the NYN domain of KHNYN reduce its restriction ability, similar to others [21]. We also show that mutations in the KH-domain of KHNYN reduces its restriction ability [53]. Recent structural and biochemical studies showed the extended di-KH domain of KHNYN does not bind RNA but the full-length KHNYN protein binds RNA without CpG preferences and the NYN domain exhibits single-stranded endonuclease specificity [60]. Furthermore, how these preferences and activity change over evolutionary time is unclear. Finally, while we identified serendipitously the potent platypus KHNYN restriction factor, its mechanism of action is unclear. We speculate that this potent ancestral KHNYN required attenuation for the genesis of retroelement-derived NYNRIN and retroviral-associated placentation acquired during the rapid evolutionary leap from monotremes to therian mammals between 180 and 160 million years ago [34,95].

## Limitations

We appreciate limitations exist in this study attempting to synthesize and understand the multiple (and sometimes conflicting) reports regarding CpG-depletion in cytoplasmic-resident RNAs and viral adaptation driven by ZAP, KHNYN, and related antiviral genes. Throughout this study, we primarily use a common laboratory strain of IAV and a previously published CpG-enriched segment 5 mutant strain (PR8 and PR8$_{CG}$) as well as in some assays an avian strain (OH175) and human pandemic strain (Cal04). Future work will benefit from examining ZAP, KHNYN, and TRIM25 susceptibility of many influenza strains derived from humans, wild birds, agricultural poultry, and domestic mammals as well as emerging mammalian H5N1 IAV isolates [96]. This work largely uses chicken cell lines and relies on overexpression of many host restriction factors by transfection or stable transduction. While aspects of our work were substantiated by knockdown and/or knockout of endogenous host cell factors, future work will benefit from using primary cells [97]. Furthermore, future work will also benefit from cell systems derived from avian reservoir species including shorebirds that currently do not exist as tractable laboratory models. We attempted to build chicken DF-1 cells constitutively expressing both ZAP and KHNYN using native human CDS. Future work may benefit from codon-optimized and inducible expression systems compared across different host species. Finally, we acknowledge two aspects of KHNYN antiviral activity shown here are limited.

Specifically, the mechanisms of KHNYN antiviral activity should be fully elucidated in the future including potential cofactors. Curiously, we found platypus KHNYN potently active in chicken DF-1 cells and human HEK293T cells suggesting either a conserved cofactor or cell-autonomous restriction factor activity. Many restriction factors exhibit cell-autonomous activity. Human and mouse APOBEC3 proteins function almost universally in vertebrate cell lines against retroviruses as well as in yeast against many Ty3 retrotransposons [98]. Similarly, human TETHERIN restricts avian leukosis viruses when expressed in chicken cells and chicken TETHERIN restricts HIV-1 when expressed in human cells [99]. Future work should examine what cofactors, proteins, and/or RNA molecules interact [100] with mammalian and avian ZAP as well as mammalian KHNYN across different cellular contexts and their impact on antiviral activity. In addition, the potent antiviral platypus KHNYN orthologue is derived from a predicted mRNA and protein product and investigation of its mechanism of action will be vastly informative. However, the sum of the results shown here should spur multiple important research directions and warrant intense interest in multiple fields including host-pathogen interactions, virus evolution, mammalian evolution, and embryonic development.

## Methods

### Cell lines

Mammalian cells, chicken DF-1 cells, and duck CCL-141 cells were grown in DMEM supplemented with 10% fetal bovine serum (FBS) and 1% penicillin/streptomycin. Chicken DT40 cells were grown in DMEM supplemented with 10% FBS, 10% tryptose phosphate buffer, 1% penicillin/streptomycin, and 50 μM beta-mercaptomethanol. Cells were maintained at 37 °C, 5% $CO_2$, and 50% humidity. HEK293T (male human epithelial kidney, ATCC, CRL-3216), A549 (male human lung carcinoma, ATCC, CCL-185), PK-15 (male porcine kidney, ATCC, CCL-33), MDCK (female canine kidney, ATCC, CCL-34), DF-1 (unspecified sex chicken embryonic fibroblast, ATCC, CRL-12203), DT40 (unspecified sex chicken lymphoblast, ATCC, CRL-2111), and CCL-141 (unspecified sex duck embryonic fibroblast, ATCC) cells were obtained from ATCC and used without extensive passaging. Sup-T11 (male human T lymphoblast clone derived from Sup-T1, ATCC, and CRL-1942) cells have been described previously [101].

### Plasmids

A CpG-enriched IAV segment 5 (encoding nucleoprotein, NP) derived from Gaunt and colleagues [47], ordered from IDT as a gBlock, and cloned into a pDZ bidirectional influenza vRNA/mRNA expression plasmid [102] via In-Fusion (Takara Bio). Expression plasmids for human ZAP-L, ZAP-S, KHNYN, and TRIM25 as well as pig ZAP, chicken ZAP, and pig KHNYN were generated using conventional molecular biology techniques. Specifically, RNA was extracted from human A549 cells, chicken DF-1, or pig PK-15 cells using Qiagen RNeasy Mini Kit, cDNA synthesized using QIAGEN Quantitect cDNA Synthesis Kit, and genes of interest were amplified by PCR from first-strand cDNA, restriction enzyme digested, and ligated into bespoke MLV-based retroviral vectors also harboring blasticidin S deaminase (BSD) or hygromycin resistance genes downstream of an IRES for bicistronic expression [100,103,104] or bespoke pcDNA3-based expression vectors. Dog and platypus KHNYN as well as dog and platypus ZAP were ordered from IDT as gBlocks. Duck ZAP was synthesized by Twist Biosciences followed by subcloning into additional plasmid backbones. Mutants, chimeras, and fluorescently tagged versions of all plasmids were generated by site-directed mutagenesis and/or overlapping PCR and subcloned into expression vectors using Phusion DNA polymerase and T4 DNA ligase (NEB). RCAS-GFP plasmid [105] was a gift of Connie Cepko (Addgene #13878). ROSV-mCherry was generated by replacing EGFP in RCAS-GFP with mCherry followed by site-directed mutagenesis to inactivate the Env glycoprotein open reading reframe. HIV-mCherry has been described previously [87]. HIV$_{CG}$-mCh was cloned by inserting a gBlock (IDT) encoding env86-561CpG [21] (from NCBI: MN685350.1 [21]) into an HIV-1 NL4-3 Env- Vpr- Nef- mCherry+ (HIV-mCh) reporter virus expression plasmid. MLV-Gag-HA-mCh was a kind gift of Nathan Sherer [106]. Codon-optimized HIV-1 NL4-3 Env (SynEnv) was synthesized

as a gBlock by IDT prior to subcloning into pcDNA3.2. pCAG-RABV-G (Addgene #36398) expressing Rabies virus glycoprotein (RABV-G) was a kind gift of Connie Cepko [107]. CRISPR/Cas9 editing of human A549 cells was performed by lentiviral transduction using pLentiCRISPR1000 [108]. Attempts at genetic editing in avian cells with pLentiCRISPR1000 were unsuccessful. CRISPR/Cas9 editing of avian cells was performed by sequential retroviral transduction of Cas9-GFP expression vector followed by single guide RNA (sgRNA) expression vector—both bespoke plasmids generated for this study. Briefly, guide RNAs were designed from the literature, Synthego design tool, or RGEN Cas-Designer [109], complementary oligos with Esp3I restriction enzyme compatible ends were annealed, and subsequently ligated into pLentiCRISPR1000 or pMLV.miRFP670.iU6sgRNA via Golden-Gate ligation with T4 DNA ligase. IAV bidirectional rescue plasmids (pDZ) have been described previously [110]. Lentiviral constructs for shRNA knockdown of chicken ZAP were cloned by annealing complementary oligos with AgeI and EcoRI restriction enzyme compatible ends and subsequently ligated into pLKO.2 via Golden-Gate ligation with T4 DNA ligase. shRNAs were designed targeting chicken ZC3HAV1 using the Broad Institute's Genetic Perturbation Platform [111]. DNA constructs were sequence confirmed by whole plasmid sequencing performed by Plasmidsaurus using Oxford Nanopore Technology or Sanger sequencing performed by Eurofins Scientific or GeneWiz from Azenta. shRNA, siRNA, and sgRNA sequences are provided in S1 Table.

## Retroviral transductions and stable cell generation

Simple retroviruses for transduction were generated by transfecting HEK293T cells with 1 μg vector plasmid [100,104], 1 μg pMD-gagpol [112], and 200 ng pMD-VSVG [113] using Transit-LT1 (Mirus Bio), changing media after 24 hours, and harvesting culture supernatant at 48 hours post-transfection followed by 0.45 μm syringe filtration. Lentiviruses for transduction were generated by transfecting HEK293T with 1.25 μg pLentiCRISPR1000 or pLKO.2_shRNA plasmid, 750 ng psPAX2 (Addgene plasmid #12260; encoding Gag/GagPol, Rev, Tat), and 200 ng pMD-VSVG, changing media after 24 hours, and harvesting culture supernatant at 48 hours post-transfection followed by 0.45 μm syringe filtration. Transducing virus preps were aliquoted and stored at −20 °C. Stable cells were generated as previously described [87,114]. Briefly, ~2,500 target cells were seeded into a 96-well flat bottom plate, allowed to adhere overnight, and 20–200 μL of transducing viral supernatant with up to 10 μg/mL polybrene added to each well. After 2–3 days in culture, transduced cells are washed with PBS, and fresh media containing antibiotic (2 μg/mL puromycin, 2 μg/mL blasticidin, or 200 μg/mL hygromycin; GoldBio, Inc) added. Cells were grown and expanded in the ongoing presence of antibiotic. In the case of CRISPR/Cas9 lentiviral transduction, A549 cells were single cell cloned by serial dilution (re-plating from pooled transducing culture as above into 96-well flat bottom plates at 25, 5, 1, and 0.2 cells per well or by gradient dilution down and across 96-well plates). After 2–4 days, single cell-derived colonies were marked and followed regularly, expanded, and then screened initially by western blot or PCR. Multiple independent clones were screened for each knockout. Genetic knockout in A549 cells was confirmed by western blot or genomic DNA extraction, amplification of Cas9 target site, blunt cloning into pJET1.2 (Thermo Scientific), and Sanger sequencing multiple transformed clones. Double knockout cells were transduced simultaneously (A549.ZK1 and A549.ZK2) with ZAP-targeting (blasticidin-resistant) and KHNYN-targeting (puromycin-resistant) CRISPR/Cas9 lentiviruses, selected with both antibiotics after 48 hours, single cell cloned, screened by western blot or PCR, and confirmed by Sanger sequencing. Avian cells were transduced with simple retrovirus Cas9-GFP (puromycin-resistant). Avian cell knockouts were maintained as pools stably expressing Cas9 and sgRNAs (blasticidin-resistant). Genetic knockout in avian cells was determined by genomic DNA extraction, PCR amplification of Cas9 target site, Sanger sequencing of pooled amplicons, and computational deconvolution of overlapping chromatograms by TIDE analysis [115].

## Influenza viruses

Wild-type Influenza A/Puerto Rico/8/1934(H1N1) (PR8) was rescued by plasmid-based transfection into HEK293T cells using Lipofectamine 3000 (ThermoFisher Scientific) and amplified in 10-day old embryonated chicken eggs. PR8$_{CG}$ virus was made by plasmid-based transfection with the WT segment 5 plasmid replaced with CpG-enriched segment 5 and

amplified in embryonated chicken eggs. Single-cycle PR8 and Cal04 (A/California/04/2009(H1N1)) that lack virus-encoded HA and instead express mCherry from the HA segment were rescued by plasmid-based transfection into HEK293T cells and amplified in MDCK cells stably expressing PR8 HA [116]. Chimeric OH175 Influenza A/Green-Winged Teal/Ohio/175/1986(H2N1) viruses [67] were rescued by plasmid-based transfection into HEK293T cells using six internal segment plasmids from OH175 as well as the HA and NA segment plasmids from PR8 for biosafety reasons and subsequently amplified in MDCK cells. PR8, PR8$_{CG}$, and OH175 transfections were performed in combination with pCAGGS expression plasmids as needed. OH175 plasmids (also known as S009) were a kind gift of Andrew Mehle. Viral titers were determined by plaque assay in MDCK cells as previously described [117].

## Influenza multicycle replication assays

For multicycle IAV infections, 500,000 A549 or DF-1 cells were plated in 6-well plates (or 100,000 CCL-141 in 12-well) and allowed to grow overnight for up to 24 hours. Cells were infected at an MOI of 0.05 infecting a confluent culture of cells, as previously described [117]. For each cell condition (e.g., DF-1 Vector, ZAP, and KHNYN) and for each virus (e.g., PR8 or PR8$_{CG}$) there were three independently infected well replicates (with the exception of multicycle replication assays in S6A Fig using two independently infected well replicates). Briefly, cells were washed with PBS prior to addition of 1mL infection media (1× PBS, 2.5% bovine serum albumin, 1% calcium/magnesium). Viral stocks were diluted such that 40 μL of inoculum could be added to each well yielding an MOI of 0.05 plaque-forming units per cell (pfu/cell). After the addition of viral inoculum, cells were incubated at 37 °C for 1 hour. Next, cells were washed with PBS prior to addition of 2.5mL viral growth medium (1x DMEM, 2.5% HEPES buffer, 2.5% bovine serum fraction V [7.5%], 1% penicillin/streptomycin) supplemented with TPCK-resistant trypsin (1 μg/mL for A549 as well as 0.625 μg/mL for DF-1 and CCL-141). Culture supernatants (at least two aliquots of 200 μL each) were collected immediately after addition to wells (0 hours post-infection) and at 24- and 48-hours post-infection (hpi) and optionally at 12-hours post-infection. Supernatants were stored at −80 °C until viral titer was determined by plaque assay in MDCK cells as previously described [117]. Multicycle replication assays were performed in triplicate with key results being replicated in independent assays (e.g., ZAP-S relative to Vector in Fig 2A, 2B, and 2C). Plaque assays were performed and quantified single-blinded. For transient knockdown by siRNA, 100,000 CCL-141 cells were plate in 12-well plates and transfected with mock, 5nM scramble control, and 1nM or 10nM pools of 10 duplex siRNAs targeting *Anas platyrhynchus* ZC3HAV1 (aplZAP) using RNAiMAX (ThermoFisher). siRNAs were designed using IDT dsiRNA design tool and sequences are provided in S1 Table.

## Single-cycle retrovirus assembly and infectivity assay

For ROSV, 500,000 DF-1 cells were plated into 6-well plates in 2mL media, cultured overnight, and transfected with 1.6 μg RCAS-mCherry, 200ng pMD-VSVG, 100/200/400ng gene of interest plasmid, and topped up to 2.2 μg total plasmid DNA mass with pBlueScript using TransIT-LT1 (Mirus Bio). At 24 hours post-transfection, we exchanged media. At 48 hours post-transfection, virus-containing supernatants were harvested and filtered using 3mL syringes and 0.45 μm syringe PVDF filters. Transfections were performed as dose-titrations of gene of interest plasmid amounts rather than triplicates of a single plasmid amount. Culture supernatants were used to infect chicken DT40 suspension cells in triplicate and the remaining volume was stored at −80 °C. We performed flow cytometry to detect infected (i.e., % mCherry-positive) DT40 cells at 48 hours post-infection using a Becton Dickinson FACS Canto II with a high-throughput plate-based sample aspirator. For MLV, HEK293T cells were plated and transfected with 500ng pMD-gagpol, 1,100ng pMLV-Gag-HA-mCherry, 200ng pMD-VSVG, 100/200/400ng gene of interest plasmid, and topped up to 2.2 μg total plasmid DNA mass with pBlueScript. For HIV-1, HEK293T cells were plated and transfected with 1,600ng pHIV-mCherry, 200ng glycoprotein-expression plasmid (pMD2-VSVG, pCAG-RABV-G, or pcDNA.HIV.SynEnv), 100/200/400ng gene of interest plasmid, and topped up to 2.2 μg total plasmid DNA mass with pBlueScript. MLV single-cycle infectivity was detected by infecting HEK239T cells and HIV-1 infectivity was detected using SupT11 cells (for VSV-G and HIV Env pseudo-typed virus) or HEK293T cells (for RABV-G pseudo-typed virus).

## Western blotting and antibodies

A549 cells were treated with 1000 U/mL of universal interferon-alpha (PBL Assay Sciences, #11200-2) for 24 hours prior to pelleting cells and lysing in 2.5× reducing sample buffer (RSB; 125 mM Tris-HCl, 20% glycerol, 7.5% SDS, 5% β-mercaptoethanol, 250 mM DTT, 0.05% Orange G, pH 6.8), and boiled at 98 °C for 15 min prior to storage at −80 °C. Transfected DF-1 and HEK293T cells for single-cycle retrovirus assembly assays were harvested at 48 hours post-transfections, pelleted by centrifugation, lysed in 2.5× reducing sample buffer (RSB), and boiled at 98 °C for 15 min prior to storage at −80 °C. Virus-containing supernatants (1 mL) from retrovirus assays were underlaid with 200 μL of 20% sucrose (diluted in 1× PBS), centrifugated at ≥21,500$g$ for two hours at 4 °C, liquid aspirated, and pellet resuspended in 2.5× RSB. Western blots were performed conventionally using commercial precast polyacrylamide gels (26-well Bio-Rad, 4%–20% Criterion TGX, #5671095) for electrophoresis (60 volts for 60 min followed by 120 volts for 60 min) and transferred to 0.2 μm nitrocellulose membrane using Invitrogen Power Blotter XL (ThermoFisher Scientific; ≤25 volts). Membranes were washed in PBS and PBST (PBS with 0.1% Tween-20) followed by blocking (2% dry milk in PBST). With constant rocking at all steps, membranes were incubated with primary antibodies overnight at 4 °C, followed by three washes for 5 min each with PBST, incubation with secondary antibodies in 2% milk in PBST for at least 1 hour at room temperature, followed by three washes for 5 min each in PBST, and one final wash in PBS. Primary antibodies used at indicated dilutions are provided in S1 Table. Secondary antibodies used include goat anti-mouse IgG IRDye 680LT and goat anti-rabbit IgG IRDye 800CW (both used at 1:10,000 dilution in 2% milk in PBST). Membranes were imaged using a LiCor Odyssey Fc infrared fluorescent imager for 2 min or 10 min per channel.

## GFP-Trap immunoprecipitation

Chicken DF-1 cells including those stably expressing YFP-tagged ZAP and KHNYN orthologues were infected with PR8 and PR8$_{CG}$ at MOI = 5 as described above in 10 cm dishes and harvested at 24 hours post-infection. A portion of each sample set aside for 10% input RNA (with RNase Inhibitor) and 10% input protein (resuspended in RSB and boiled), then stored at −80 °C. Remaining cells were washed twice with cold 1× PBS (containing RNase inhibitor) and resuspended in radioimmunoprecipitation assay buffer (RIPA: 10 mM Tris-HCl, 150 mM NaCl, 0.5 mM EDTA, 0.1% SDS, 1% Triton-X, 1% sodium deoxycholate, 2.5 mM MgCl$_2$, 1 mM phenylmethylsulfonyl fluoride, 1 unit/mL DNaseI, and 1 unit/mL RNase Inhibitor (Roche, #3335399001). Samples were needle-lysed with a 26G needle, slowly inverted continuously at 4 °C for 30 min, and centrifugated at ≥21,500$g$ for 10 min at 4 °C. GFP-Trap agarose beads (ProteinTech, #gta) were equilibrated in dilution buffer (10 mM Tris-HCl, 150 mM NaCl, 0.5 mM EDTA) with 50 μL bead slurry per sample into 500 μL dilution buffer per sample. Beads were centrifugated at 2,500$g$ for 5 min at 4 °C and washed with dilution buffer twice more. Remaining RIPA supernatant of each sample was added to 500 μL of equilibrated beads in dilution buffer and slowly inverted continuously at 4 °C overnight. Beads were centrifugated at 2,500$g$ for 5 min at 4 °C and resuspended in wash buffer (10 mM Tris-HCl, 150 mM NaCl, 0.5 mM EDTA, 0.05% IGEPAL) twice and then split into two equivalent volume samples. One subjected to RNA extraction by Qiagen RNeasy Kit. The other centrifugated once more, supernatant aspirated, bead pellet resuspended in RSB, boiled at 98 °C for 15 min, and stored at −80 °C.

## Fluorescence microscopy

Fixed-cell images were obtained on a BioTek Cytation 5 (Agilent) using a 4× (NA = 0.13, #1220519), 20× (NA = 0.40, #1220517), or 40× (NA = 0.6, #1220544) Plan Fluorite objective and LED-filter cube sets for DAPI (excitation = 377 ± 50 nm; emission = 447 ± 60 nm; #1225007 and #1225100), GFP for mNeonGreen (excitation = 469 ± 35; emission = 525 ± 39; #1225001 and #1225101), and TRITC for mCherry and AlexaFluor568 (excitation = 556 ± 29; emission = 600 ± 37; #1225012 and #1225125). We quantified single-cycle IAV infections by imaging and used the BioTek Gen5 software to calculate percent infectivity using scPR8 (MOI = 5) and scCal04 (MOI = 5) encoded mCherry as a readout for

infectivity (# TRITC+ cells/ # DAPI-stained nuclei). To quantify OH175 nonreplicating infections, we performed immuno-fluorescence on infected cells (MOI = 0.5 without TPCK trypsin to block virus transmission) fixed with 4% paraformalde-hyde at 24 hours post-infection. Briefly, we subjected cells to permeabilization (PBS with 0.25% Triton-X-100) for 10 min, blocking (PBS with 3% bovine serum albumin), primary antibody staining with mouse anti-NP (BEI, NR-43899, 1:500) in blocking buffer (3% bovine serum albumin) for 1 hour, secondary antibody staining with goat anti-rabbit AlexaFlour-568 (Invitrogen, #A-11031, 1:500), and nuclear counterstaining with DAPI (Sigma, D9542, 100 ng/mL).

### Minigenome reporter assay

Chicken DF-1 cells were seeded into 96-well glass-bottom plates, allowed to adhere overnight, and then transfected with equal amounts of plasmids encoding IAV heterotrimeric polymerase (pCAGGS.PB2, pCAGGS.PB1, and pCAGGS.PA), wild-type IAV NP (pCAGGS.NP), and a vRNA/negative-sense IAV-codon-optimized mCherry expression plasmid (pPol. mCherry) [118] as well as plasmids encoding mNeongreen or mNeongreen-tagged mammalian KHNYN orthologues with TransIT-LT1. At 24 hours, cells were washed with 1× PBS, fixed with 4% paraformaldehyde, counterstained with DAPI, and imaged for mNeongreen and mCherry expression as a % mCherry-positive as described above.

### Structural models of ZAP and KHNYN

For structural models of ZAP, amino acid sequences for the four zinc finger domains of different species and mutants of ZAP were used as input for RobettaCM using the human ZAP crystal structure (PDB: 6UEJ) for comparative modeling. We noted that in most cases, ab initio folding by AlphaFold or RoseTTAFold may not capture metal ion-coordinated finger structures. Sequences used are provided in S1 Table. Models generated and 6UEJ are shown in S1 Data. For structural models of KHNYN, amino acid sequences for the KH domain or NYN domain of KHNYN were used as input for RoseT-TAFold using default settings. Sequences used are shown in S1 Table. Both RoseTTAFold and RobettaCM are publicly available at robetta.bakerlab.org. All models generated are provided in S2 and S3 Data. Models were visualized in Chi-mera (v1.13.1, build 41965, UCSC) and aligned using MatchMaker with a Needleman-Wunsch alignment algorithm and BLOSUM-62 matrix.

### Phylogenetic analyses

Coding sequences (CDS) of human KHNYN, NYNRIN, N4BP1, and their orthologues were downloaded from ensembl. CDS of ZAP homologs were downloaded from ensembl. CDS were aligned using ClustalOmega in Seaview (V5.0.4). Phylogenetic trees were generated in Seaview using PhyML with 100 bootstraps using CDS. Synteny schematics were generated manually in Adobe Illustrator using data from UCSC ENCODE Genome Browser, NCBI, and ensembl. Align-ments and phylogenetic trees are available in S4 and S5 Data.

### Virus bioinformatic analysis

Full-length influenza virus sequences were downloaded from the Influenza Research Database for each animal host species (human A, avian A, swine A, human B, and human C), including the nucleotide sequence, host, collection date, and subtype. Dinucleotide values were counted by string count. Data were plotted using ggplot2. Data are available at a zenodo record (https://doi.org/10.5281/zenodo.14919910) and R code is available in S1 File. Sliding window CpG data were analyzed using an in-house custom script (cpg_counts.R) available in S1 File.

### RNA-seq

RNA was extracted from mock and infected samples at 24 hours post-infection (MOI = 0.05) using Qiagen RNeasy Kit. Samples were submitted for polyA-selected RNA (mRNA) sequencing using an Illumina NovaSeq 6000 with 150 bp paired

end reads at the University of Illinois Urbana-Champaign Roy J. Carver Biotechnology Center. Reads were trimmed of adapter sequences and low-quality reads eliminated using Trimmomatic (V0.32). Trimmed reads were mapped to a hybrid reference genome (Gallus gallus GRCg7b, PR8, and PR8$_{CG}$) using STAR (V2.7.11b) and quantified per feature using HTSeq (V2.0). Differential expression analysis was performed using DESeq2 (v1.34.0). Data were processed and analyzed in R-Studio version 1.3.1073 using R language version 3.6.3. DESeq2 R code is available in S2 File. RNA-seq data are available at the NCBI Sequence Read Archive (SRA) under accession code PRJNA1208514.

### Quantitative reverse-transcription PCR

To quantify endogenous chicken *ZC3HAV1* and *TUBA4A* mRNA transcripts in shRNA-expressing stable cells following chicken interferon-alpha treatment (1000 U/mL; Bio-Rad) for 24 hours, 500,000 cells were plated in 6-well plates and allowed to adhere overnight in culture media followed by addition of chicken interferon or vehicle. After 24 hours, cells were harvested, RNA extracted using Qiagen RNeasy kit, and cDNA synthesized used Qiagen Quantitect Reverse-Transcription Kit using supplied primer mix (includes random hexamers and oligo-dT primers) with genomic DNA wipeout according to manufacturer's instructions. To quantify *IAV segment 5 RNA* in mock and infected chicken DF-1 cells (input) and GFP-Trap immunoprecipitated (IP) samples, RNA extraction and cDNA synthesis were performed as above. Quantitative PCR was performed in technical triplicate on a LightCycler 480 II (Roche Life Science) or QuantStudio5 (Applied Biosystems) using Ssofast Eva Green Supermix (Bio-Rad). Oligonucleotide primer sequences are provided in S1 Table.

### Statistical analyses

Plaque assay results from IAV infections were plotted in GraphPad Prism 10 and one-way ANOVA was used at 48 hours post-infection or two-way ANOVA for multiple time points to determine statistical significance each with Dunnett's test for correcting for multiple comparisons. Single-cycle ROSV and IAV infectivity results were plotted in GraphPad Prism 10. Unpaired *t* tests were used to calculate statistical significance for IAV single-cycle infectivity assays with Holm-Šidák's test correcting for multiple comparisons. Mean and standard deviation of the mean were calculated in GraphPad Prism 10. qPCR calculations were performed in Microsoft Excel. Statistical analyses and data values underlying graphs are available in S2 Table or in a data repository at https://doi.org/10.5281/zenodo.14919910.

## Supporting information

**S1 Fig. Mammalian IAV sequences are CpG-depleted over time relative to avian IAV. (A)** IAV sequences from human (blue) and swine (green) hosts exhibit depletion of CpG content (CpG per kb = # CpG/segment length in kilobases (kb)) relative to avian (orange) IAV over time since 1918. IBV (pink) and ICV (purple) exhibit relatively stable but dramatically lower CpG content than all IAV segments. Thick lines represent linear trend of data surrounded by 95% confidence interval in lighter shading. **(B)** CpG content of IAV sequences from avian, human, and swine segregated by virus segment. Segments 1–3 include PB2, PB1, and PA encoding the viral Polymerase. Segments 4–6 include HA, NP, and NA encoding virion proteins often subject to adaptive immune pressure. Segments 7–8 include M and NS. **(C)** CpG content of IAV sequences from avian, human, and swine hosts segregated by common HA and NA subtypes. **(D)** CpG content of IAV sequences segregated by time periods indicated on the x-axis (top). Schematic of IAV natural history showing avian H1N1, H2N1, and H3N2 in orange shaded lines, human circulating subtypes in shaded blue lines, 2009 H1N1 pandemic swine flu in green lines, (middle) and source of reassortant segments in matched colors (bottom). Thick black bars represent median CpG content, boxes define the 25–75th percentile, and whiskers represent range excluding outliers. Horizontal dashed gray lines highlight median avian IAV CpG content as a reference. The data underlying this figure and code used to generate the subpanels can be found in https://doi.org/10.5281/zenodo.14919910.
(EPS)

**S2 Fig. Dinucleotide content of influenza viruses. (A)** CpG content as the normalized rho ($\rho$ = #CpG/length divided by the product of #C/length and #G per length) of all IAV sequences from avian, human, and swine hosts as well as IBV and ICV. **(B)** Percent GC content (%GC = (#G + #C)/ total length) all IAV sequences from avian, human, and swine hosts as well as IBV and ICV. **(C)** GpC content of all IAV sequences from avian, human, and swine hosts as well as IBV and ICV. **(D)** Remaining dinucleotide content of all IAV sequences from avian, human, and swine hosts as well as IBV and ICV. The data underlying this figure and code used to generate the subpanels can be found in https://doi.org/10.5281/zenodo.14919910.
(EPS)

**S3 Fig. Structural comparison of avian ZAP proteins and data for avian ZAP expression and knockdowns.**
**(A)** Structure and model of human (blue; PDB: 6UEJ), chicken (RobettaCM, orange), chicken (RoseTTAFold, medium orange), and chicken (AlphaFold2, light orange) ZAP zinc fingers (1−4) with a CpGpU RNA oligonucleotide (gray; PDB: 6UEJ). Structure and model of human (blue; PDB: 6UEJ), duck (RobettaCM, green), duck (RoseTTAFold, medium green), and duck (AlphaFold2, light green) ZAP zinc fingers (1−4) with a CpGpU RNA oligonucleotide (gray; PDB: 6UEJ). Structure and model of human (blue; PDB: 6UEJ), quail (RobettaCM, pink), quail (RoseTTAFold, medium pink), and quail (AlphaFold2, light pink) ZAP zinc fingers (1−4) with a CpGpU RNA oligonucleotide (gray; PDB: 6UEJ). Structure and model of human (blue; PDB: 6UEJ), chicken (RobettaCM, orange), duck (RobettaCM, green), and chicken (AlphaFold2, light pink) ZAP zinc fingers (1−4) with a CpGpU RNA oligonucleotide (gray; PDB: 6UEJ). **(B)** Root mean squared deviation (RMSD) of structures and models in (A). The models underlying these subpanels in (A and B) can be found in S1 Data. **(C)** Western blots of chicken DF-1 cells stably expressing mNG, mNG-ggaZAP, and untagged ggaZAP as well as Vector transduced cells detecting mNG and blasticidin S deaminase (BSD). Actin detected as a loading control. **(D)** Western blots of chicken DF-1 cells stably expressing mNG, mNG-ggaZAP, and mutants of mNG-ggaZAP as well as Vector transduced cells detecting mNG and BSD. Actin detected as a loading control. **(E)** qPCR results detecting endogenous gga *ZC3HAV1* mRNA in chicken DF-1 cells transduced with lentiviruses expressing shRNAs indicated (shades of gray) following treatment with 1,000 U/mL of chicken interferon-alpha (IFN) relative to mock treated cells normalized to gga *TUBA1A* using a $2^{-ddCt}$ calculation. **(F)** Titers of PR8 and PR8$_{CG}$ over 48 hpi in chicken DF-1 cells stably expressing shRNAs targeting endogenous ggaZAP (shggaZAP1-3, shades of gray triangles/diamonds/hexagons), a control scrambled sequence siRNA (shScramble, dark gray squares), and parental untransduced DF-1 cells (black circles). MOI = 0.05. Individual data points are displayed as 50% transparent jittered shapes behind mean data point with error bars representing standard deviation of the mean. **(G)** Titers of PR8 and PR8$_{CG}$ at 48 hpi in duck CCL-141 cells transiently transfected with 1 nM or 10 nM siRNAs targeting endogenous aplZAP (siZAP, shades of yellow), a control scrambled sequence siRNA (siNeg, 5 nM, gray), and parental untransfected (black) CCL-141 cells. Bar represents mean PFU/mL with error bars representing standard deviation of the mean. Individual data points are displayed as black circle outlines. **(H)** Western blots of chicken DF-1 cells stably expressing human ZAP-L, ZAP-S, KHNYN, and TRIM25 as well as dually transduced ZAP-L or ZAP-S with KHNYN or dKHNYN including Vector transduced cells detecting ZAP and KHNYN. GAPDH detected as a loading control. The data underlying the graphs in this figure can be found in S2 Table. Original annotated immunoblot images can be found in S1 Raw Images.
(EPS)

**S4 Fig. TIDE analysis of pooled CRISPR/Cas9 chicken DF-1 and duck CCL-141 ZC3HAV1/ZAP knockouts. (A and B)** Schematic depicting exon organization of chicken ZC3HAV1/ZAP (orange) and duck ZC3HAV1/ZAP (yellow) as well as sites targeted by CRISPR single guide RNAs (sgRNAs; scissors). Colored boxes indicate coding exons and white boxes indicate noncoding portions of exons. Genomic sequence targeted by sgRNAs (underlined) and protospacer adjacent motif (PAM, lowercase) and amino acids encoding by that region (bold residues overlap sgRNA target). Genomic distances are relative and total locus size indicated. **(C)** Total knockout efficiency as determined by tracking of indels by

decomposition (TIDE) analysis (% Knockout) in each pool relative to LacZ control cells. **(D–F)** TIDE analysis showing percent of sequences in pool exhibiting indicated indel (−10 to +10 bp from expected cut site) in chicken ZC3HAV1-targeted KO DF-1 cells. **(G–I)** TIDE analysis showing percent of sequences in pool exhibiting indicated indel (−10 to +10 bp from expected cut site) in duck ZC3HAV1-targeted KO CCL-141 cells. Red bars indicate disruptive indels (CRISPR-mediated insertions or deletions resulting in out-of-frame mutations). Gray bars indicate un-edited sequences. Black bars indicate in-frame indels (CRISPR-mediated insertions or deletions resulting in amino acid deletions). The data underlying the graphs in this figure can be found in S2 Table.
(EPS)

**S5 Fig. Genetic and genomic comparisons of KHNYN genes. (A)** Synteny diagrams of KHNYN genomic organization in representative species. KHNYN gene in blue, NYNRIN in magenta, neighboring human/pig/koala genes in black, neighboring platypus genes in red, and human/chicken genes syntenic to red genes in gray. Distances are relative and approximate size of locus shown on the right in kilobasepairs (Kbp). **(B)** Schematic of exon organization (left) and protein domain architecture (right) of human N4BP1 (pink), KHNYN (blue), and NYNRIN (magenta). KH domain represents the extended HNRNPK-like, GXXG-like motif residues typical of KH-domains are indicated, NYN represents the NYN/PIN endonuclease domain, and CU represents cullin-binding domain associated with NEDD8 (KHNYN and NYNRIN) or cousin of CUBAN domain (N4BP1). **(C)** Maximum likelihood phylogeny of all ensembl-annotated orthologues of human KHNYN, NYNRIN, and N4BP1. **(D)** Synteny diagrams and schematics of exon organization for alligator, anole, and zebrafish *"khnyn-like"* genes. **(E)** Structural predictions in three views of human (dark blue), pig (medium blue), dog (cyan), and platypus (lime green) KHNYN (residues 1–213). Putative GXXG motif highlighted by dashed black circle. Platypus-specific short loop highlighted by dashed black square. The data underlying this figure can be found in S2 and S4 Data.
(EPS)

**S6 Fig. YFP-tagged human ZAP-S and KHNYN inhibit PR8$_{CG}$ and OH175 IAV. (A)** Titers of PR8, PR8$_{CG}$, and avian OH175 at 48 hpi in chicken DF-1 cells stably expressing YFP-tagged human ZAP-L (orange), human ZAP-S (green), chicken ggaZAP (orange/gray), KHNYN (blue), dKHNYN (gray) and platypus oanKHNYN (pale blue) relative to YFP-alone (black). MOI = 0.05. Bar represents mean PFU/mL with error bars representing standard deviation of the mean. Individual data points are displayed as black circle outlines. **(B)** Western blots of chicken DF-1 cells used in (A) detecting YFP. Histone H3 detected as a loading control. **(C)** CpG content of OH175 (shades of red by segment length), OH175 segment 3 PA bolded line, as well as PR8 segment 3 PA (bold blue line) calculated as # CpG per 100 nucleotides (nt) over a sliding window (top). Schematic of PR8 and OH175 viruses and displaying CpG content for each virus and total CpG number encoded by each PA segment (bottom). **(D)** Fold enrichment by qPCR for PR8 IAV segment 5 NP RNA following GFP-Trap immunoprecipitation of indicated YFP-tagged proteins from chicken DF-1 cells uninfected or infected with PR8 or PR8$_{CG}$. MOI = 5. Fold enrichment is shown relative to parental chicken DF-1 cells lacking YFP, similar to an IgG isotype control used in chromatin immunoprecipitation (ChIP-qPCR) using a $2^{-ddCt}$ calculation. Western blots detecting YFP in IP (top) and YFP and IAV NP in Input (bottom). Actin detected as loading control. Individual data points are displayed as black circle outlines. The data underlying the graphs in this figure can be found in S2 Table. Original annotated immunoblot images can be found in S1 Raw Images.
(EPS)

**S7 Fig. Chicken and IAV transcriptome largely unaffected by human ZAP or KHNYN. (A)** Principal component analysis (PCA) of RNA-seq data. **(B)** Heatmap showing differentially expressed IAV mRNAs. **(C)** Heatmap showing most differentially expressed chicken mRNAs. The data underlying this figure and code used to generate the subpanels can be found in https://doi.org/10.5281/zenodo.14919910.
(EPS)

**S8 Fig. Platypus KHNYN is a potent and broadly acting antiviral protein. (A)** Retroviral assembly and single-cycle infectivity results of mammalian ZAP-L and ZAP-S homologs including pig, dog, and platypus on ROSV infectivity (relative infectivity, %mCh+). Platypus encodes a well-annotated ZAP-L homologue but not a ZAP-S, a C653S prenylation mutant was included. **(B)** Retroviral assembly and single-cycle infectivity results of mammalian KHNYN homologs including pig, dog, and platypus as well as catalytically inactivated (*"d"*) mutants of each on ROSV infectivity. **(C)** Retroviral assembly and single-cycle infectivity results of platypus KHNYN including truncation mutants on ROSV infectivity (top). Schematic of protein domain architecture of platypus KHNYN with arrows indicating position of truncating stop mutation (bottom). **(D)** Retroviral assembly and single-cycle infectivity results of platypus KHNYN including truncation mutants on ROSV infectivity (top). Indication (bottom) of virus tested (HIV-mCh, HIV$_{CG}$-mCh, or MLV-mCh) and glycoprotein used for pseudo-typing (HIV-1 Env, VSV-G, or RABV-G). **(E)** Retroviral assembly and single-cycle infectivity results of mammalian KHNYN homologs including pig, dog, and platypus as well as catalytically inactivated (*"d"*) mutants of each on ROSV infectivity in chicken ZAP knockout DF-1 cells (ggaZAP-3 KO from Figs 1G, S4C, and S4F). Bar represents mean relative infectivity with error bars representing standard deviation of the mean. Individual data points are displayed as black circle outlines. The data underlying the graphs in this figure can be found in S2 Table. Original annotated immunoblot images can be found in S1 Raw Images.
(EPS)

**S9 Fig. ZAP localization conferred by prenylation motif in mammal and avian species. (A)** Fluorescence images of DF-1 cells expressing mNG as well as fusion proteins of mNG-ZAP orthologues and mutants indicated (green). Cell and nuclei outline in dashed white lines. Scale bars represent 10 μm. **(B)** Amino acid alignment of CaaX prenylation motif from multiple species indicated. The data underlying (**B**) can be found in S5 Data.
(EPS)

**S10 Fig. Conservation of mammalian KHNYN endonuclease domain and mammalian KHNYN subcellular localization. (A)** Amino acid alignment of NYN/PIN endonuclease domain (human residues 437−589) from multiple species KHNYN orthologues indicated. The data underlying (A) can be found within S4 Data. **(B)** Structural predictions in three views of human (dark blue), pig (medium blue), dog (cyan), and platypus (lime green) KHNYN (residues 437−589). Catalytic aspartic acid residues highlighted by dashed black circle. The models underlying (B) can be found in S3 Data. **(C)** Fluorescence images of DF-1 cells expressing mNG as well as fusion proteins of mNG-KHNYN orthologues and mutants indicated (green). Cell and nuclei outline in dashed white lines. Scale bars represent 10 μm.
(EPS)

**S1 Table. Key reagents.** Spreadsheet.
(XLSX)

**S2 Table. Statistical analyses and graphed values.** Spreadsheet.
(XLSX)

**S1 File. Influenza sequences analysis R code.** Text files.
(ZIP)

**S2 File. DEseq2 analysis R code.** Text files and DESeq2 file.
(ZIP)

**S1 Data. Human and avian ZAP structural models.** Python files open in UCSC Chimera.
(ZIP)

**S2 Data. Mammalian KHNYN structural models of KH region.** Python files open in UCSC Chimera.

(ZIP)

**S3 Data. Mammalian KHNYN structural models of NYN domain.** Python files open in UCSC Chimera.
(ZIP)

**S4 Data. N4BP1, KHNYN, and NYNRIN phylogenetic data.** Clustal nucleotide alignment, pdf of phylogenetic tree.
(ZIP)

**S5 Data. ZC3HAV1/ZAP phylogenetic data.** Clustal nucleotide alignment, pdf of phylogenetic tree.
(ZIP)

**S1 Raw Images. Original uncropped annotated immunoblot images from this study.**
(PDF)

## Acknowledgments

We wish to thank members of both Langlois and Harris labs for helpful conversations. We wish to acknowledge and thank Andrew Mehle for the A/green-wing teal/Ohio/1751986(H2N1) OH175/S009 plasmids. We thank the University of Illinois Roy J. Carver Biotechnology Center for RNA and amplicon sequencing. RCAS-GFP (CT#22) was a gift from Connie Cepko (Addgene plasmid #13878). psPAX2 was a gift from Didier Trono (Addgene plasmid #12260). pMD2.G was a gift from Didier Trono (Addgene plasmid #12259). pCAG-RABV-G was a gift from Connie Cepko (Addgene plasmid #36398). Polyclonal Anti-Influenza Virus, A/Puerto Rico/8/1934 (H1N1), (antiserum, Rooster), NR-3098 was obtained through BEI Resources, NIAID, NIH. Monoclonal Anti-Influenza A Virus Nucleoprotein (NP), Clone IC5-1B7 (produced in vitro), NR-43899 was obtained through BEI Resources, NIAID, NIH. Anti-Human Immunodeficiency Virus 1 (HIV-1) p24 Hybridoma (183-H12-5C), ARP-1513, was obtained through the NIH HIV Reagent Program, Division of AIDS, NIAID, NIH contributed by Dr. Bruce Chesebro and Dr. Hardy Chen. AMV-3C2 was deposited to the DSHB by Boettiger, D. (DSHB Hybridoma Product AMV-3C2).

## Author contributions

**Conceptualization:** Jordan T. Becker, Ryan A. Langlois.

**Data curation:** Jordan T. Becker, Frances K. Shepherd.

**Formal analysis:** Jordan T. Becker, Frances K. Shepherd.

**Funding acquisition:** Jordan T. Becker, Reuben S. Harris, Ryan A. Langlois.

**Investigation:** Jordan T. Becker, Clayton K. Mickelson, Lauren M. Pross, Autumn E. Sanders, Esther R. Vogt, Frances K. Shepherd, Chloe Wick, Alison J. Barkhymer, Stephanie L. Aron, Elizabeth J. Fay.

**Methodology:** Jordan T. Becker, Clayton K. Mickelson, Frances K. Shepherd.

**Project administration:** Jordan T. Becker, Ryan A. Langlois.

**Resources:** Jordan T. Becker, Clayton K. Mickelson, Frances K. Shepherd, Stephanie L. Aron, Elizabeth J. Fay, Reuben S. Harris, Ryan A. Langlois.

**Software:** Jordan T. Becker, Frances K. Shepherd.

**Supervision:** Jordan T. Becker, Reuben S. Harris, Ryan A. Langlois.

**Validation:** Jordan T. Becker.

**Visualization:** Jordan T. Becker.

**Writing – original draft:** Jordan T. Becker.

**Writing – review & editing:** Jordan T. Becker, Clayton K. Mickelson, Lauren M. Pross, Autumn E. Sanders, Esther R. Vogt, Frances K. Shepherd, Chloe Wick, Alison J. Barkhymer, Stephanie L. Aron, Elizabeth J. Fay, Reuben S. Harris, Ryan A. Langlois.

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
