## [Editor Report · Decision Letter 0]

24 Feb 2025

Dear Dr Becker,

Thank you for submitting your manuscript entitled "Mammalian ZAP and KHNYN independently restrict CpG-enriched avian viruses" for consideration as a Research Article by PLOS Biology.

Your manuscript has now been evaluated by the PLOS Biology editorial staff, as well as by an academic editor with relevant expertise, and I am writing to let you know that we would like to send your submission out for external peer review.

Once your full submission is complete, your paper will undergo a series of checks in preparation for peer review. After your manuscript has passed the checks it will be sent out for review. To provide the metadata for your submission, please Login to Editorial Manager (https://www.editorialmanager.com/pbiology) within two working days, i.e. by Feb 26 2025 11:59PM.

Kind regards,

Melissa

Melissa Vazquez Hernandez, Ph.D.

Associate Editor

PLOS Biology

---

## [Decision Letter · Decision Letter 1]

4 Apr 2025

Dear Dr Becker,

Thank you for your patience while your manuscript "Mammalian ZAP and KHNYN independently restrict CpG-enriched avian viruses" was peer-reviewed at PLOS Biology. It has now been evaluated by the PLOS Biology editors, an Academic Editor with relevant expertise, and by several independent reviewers.

In light of the reviews, which you will find at the end of this email, we would like to invite you to revise the work to thoroughly address the reviewers' reports. As you will see below, majority of reviewers are positive about the relevance and novelty of the study, yet some concerns have raised during revision. Reviewer 1 acknowledges the potential importance of the study but suggests that several experiments are lacking necessary controls. Reviewer 2 agrees that the study tackles an important question, but points out the absence of functional validation, specifically the lack of direct evidence for ZAP and KHNYN RNA binding/degradation, as well as a mechanistic explanation for how platypus KHNYN inhibits avian viruses. Reviewer 3 highlights that some statements in the manuscript should be toned down, particularly those based on results from lab-adapted PR8 and PR8cg strains.

Important: After discussions with the Academic Editor and the reviewers, we do not expect you to provide mechanistic insights into how platypus KHNYN functions. However, we do require you to provide biochemical or functional validation demonstrating that KHNYN can inhibit IAV independently of ZAP. Specifically, this should involve testing whether KHNYN can directly bind to or degrade RNA in the absence of ZAP expression. Additionally, we ask that you include the controls suggested by Reviewer 1 and either modify or tone down the general statements highlighted by Reviewer 3. Alternatively, testing additional IAV strains may help support such claims. Other concerns raised can be addressed through further clarification or modifications in the text.

Given the extent of revision needed, we cannot make a decision about publication until we have seen the revised manuscript and your response to the reviewers' comments. Your revised manuscript is likely to be sent for further evaluation by all or a subset of the reviewers.

**IMPORTANT - SUBMITTING YOUR REVISION**

*Re-submission Checklist*

*Published Peer Review*

*PLOS Data Policy*

*Blot and Gel Data Policy*

Sincerely,

Melissa

Melissa Vazquez Hernandez, Ph.D.

Associate Editor

PLOS Biology

REVIEWERS' COMMENTS:

Reviewer #1:

In this manuscript the authors demonstrate that avian strains of IAV contain significantly more CpG content and demonstrate avian ZAP and all iso-forms are incapable of reducing viral replication by observing this phenomenon for chicken. The authors then went on to establish several mammalian ZAP and KHNYN can independently reduce viral replication of CpG enriched IAV and ROSV by evaluating viral titers in chicken cells via multicycle replication assays. Specifically, they demonstrated that human ZAP-S and KHNYN reduce viral replication for CpG rich IAV, while ZAP-L reduced ROSV (avian virus) replication. They demonstrated that other mammalian CpG dependent restriction mechanisms (i.e platypus) can reduce virus replication and that ZAP and KHNYN can function independently. Though a mechanism was not thoroughly established, this work provides the foundation research on the independent antiviral capabilities of KHNYN and characterization of the mechanism. These studies could have important implications and impact in the field by advance our understanding on mechanisms of viral evolution, including the role of KHNYN in reducing CpG rich IAV replication, new anti-viral mechanism that is specific for CpG enriched virions in mammals, and species-specific roles of ZAP. There some minor issues with controls lacking, as described below

Major comments:

1. Figure 1D 1E and 1F show overexpression of ggaZAP with PR8 or PR8CG, and no differences are observed in titers as compared to vector or mNeogreen. The authors need to show that there is equal expression of proteins. Otherwise the comparison can not be made. In addition, a positive control needs to be included as comparison. For example human ZAP should inhibit PR8CG ? or another antiviral factor to show inhibition as control

2. The same comment about control for Figures 4C,D,E. expression of proteins transfected need to be shown. If good antibodies are not available, at least other ways to show expression should be attempted (RNA expression, tagged proteins, etc)

3. Though they do present the double knock out, ZK1 and ZK2, they should concurrently express ZAP-S and KHNYN to confirm a decrease in viral replication that is more significant than the independent phenomenon as KHNYN is a co-factor of ZAP.

Minor Comments:

1. The schematic (Figure 2A), while visually appealing, does not effectively convey a clear message.

2. Define ZK1 and ZK2 in the body of the text and/or figure legend

3. Make a note that CpG rich IAV replicates less or is somewhat attenuated compared to the original IAV strain.

Reviewer #2:

Influenza A virus (IAV) represents a significant threat to global health, given its cross-species transmission and the antiviral mechanisms that suppress IAV zoonoses are poorly understood. Previous studies have interrogated the host proviral factors involved in the cross-species interface at the amino acid and protein structure level but have not elucidated the network of restriction factors that target zoonotic IAV infections nor any viral nucleic acid adaptations driven by host-pathogen interactions during infections. Mammalian zinc finger antiviral protein (ZAP) has been shown by other groups to restrict zoonotic IAV due to ZAP's CpG-targeting activity. This study demonstrates that CpG dinucleotide suppression is a defining feature of mammalian-adapted IAV and may be driven by host restriction factors such as ZAP and KHNYN. Using comparative genomics and structural modeling, the authors show that avian IAV maintains higher CpG content, potentially due to the absence of a KHNYN orthologue and structural divergence in avian ZAP that may impair CpG recognition. Functional assays in avian cells reveal that human ZAP-S and KHNYN can independently restrict CpG-enriched IAV, while chicken ZAP fails to restrict IAV. The authors further identify platypus KHNYN as a potent and broadly active antiviral factor across multiple viruses independent of CpG content.

This study addresses an interesting and important question about why the CG dinucleotide content of avian influenza viruses is not as constrained as that of mammalian adapted influenza viruses and goes on to investigate whether ZAP and KHNYN are drivers of these viral nucleic acid adaptations. While the complementary computational, evolutionary, and structural analyses are novel and compelling, the study is limited by a lack of biochemical and functional validation, including direct evidence of ZAP and KHNYN RNA binding or degradation and mechanistic insights into how platypus KHNYN inhibits avian viruses. As a result, many of the conclusions were not supported by the experimental data and the proposed mechanisms remain speculative dampening the significance of the study.

Major comments:

1. The manuscript relies heavily on sophisticated evolutionary and structural analyses to support its conclusions but lacks biochemical validation to directly test its core hypotheses. For example, the authors attribute the inability of avian ZAP to restrict CpG-enriched IAV to divergence in ZAP's second zinc finger (ZF2), yet do not present any RNA binding assays to confirm avian ZAP cannot bind IAV as effectively as mammalian ZAP, nor that ZF2 is the main determinant of this differential binding. Furthermore, even though ZF2 contains the binding pocket for CG dinucleotide, other zinc fingers contain additional binding pockets and both the second and fourth zinc fingers are critical for ZAP's antiviral activity (Guo JVI 2004, Meagher PNAS 2019, Luo Cell Reports 2020). However, this study does not address the potential role of ZF4 and/or other zinc fingers. Similarly, this study proposes KHNYN can restrict CpG-enriched IAV in the absence of ZAP, which contradicts our current understanding of KHNYN as only a ZAP cofactor. This is a novel and interesting finding, however, no evidence is presented that KHNYN can directly bind or degrade RNA independently of ZAP expression. Experiments providing a mechanistic explanation of KHNYN restriction are essential to support this claim. In addition, it is not clear whether the endogenous ZAP protein in the chicken cells is working with overexpressed human KHNYN to inhibit CG-enriched PR8 (Figure 2B). It is also worth noting that the RNA-seq data presented show no reduction in viral mRNA levels in the context of ZAP or KHNYN, despite restriction of IAV. This raises fascinating questions about KHNYN-mediated restriction of IAV. The manuscript discusses KHNYN's mechanism of action through the lens of indirect viral mRNA degradation but does not address the possibility of alternative mechanisms.

2. There is an underexplored discrepancy in the isoform-specific restriction of different viruses by ZAP. The authors show that ZAP-S restricts IAV but not Rous sarcoma virus (ROSV), while ZAP-L restricts ROSV but not IAV. This divergence in antiviral activity is fascinating considering ZAP-L is canonically considered the more antiviral isoform of ZAP. The authors should discuss the discrepancy of their data from previous studies especially when both IAV and ROSV are CpG-enriched viruses and both ZAP isoforms contain the RNA binding domain. It would strengthen the manuscript to address isoform-specific restriction of specific viruses, particularly in the context of subcellular localization and cofactor interactions.

3. The finding that platypus KHNYN restricts a broad range of viruses is intriguing and suggests functional divergence from other mammalian KHNYN orthologs. However, the mechanism underlying this potent and broad antiviral activity remains unclear. The authors report that mutation of the NYN catalytic residues does not impair restriction, yet truncation of the endonuclease domain abolishes activity (Figure S7B-C), which is a contradiction that is not fully addressed. Furthermore, although structural modeling reveals a distinct conformation in the KH domain of platypus KHNYN, the functional implications of this difference are not explored. The study would be strengthened by additional experiments to determine whether platypus KHNYN binds viral RNA directly, whether it uses a distinct cofactor, or whether it acts through a different antiviral pathway altogether. These findings are promising but require further mechanistic resolution to clarify how platypus KHNYN achieves its broad antiviral activity.

4. Figure 1:

a. In panels D, F, G and H, there is a consistent 10-fold decrease in viral PFU/mL between the PR8 and PR8CG experiments across all conditions, including the controls. This implies that there could be some antiviral activity attributed to an unknown or unidentified host factor. This further complicates the claims made by this study, as there might exist another antiviral factor that may interact with KHNYN and/or ZAP. This observation was not discussed in the manuscript.

5. Figure 2:

a. The colors used for figures in most panels are difficult to distinguish clearly. For example, the color used for "TRIM25" and "gTRIM25" in panel A are very similar and thus was briefly confusing. Moreover, the shades of green used for ZAP-L and ZAP-S blend together and this is made more confusing by the slightly different shades of these colors used for the mutant ZAPs in panels B, D and F.

6. Figure S2:

a. The analyses in Figure S2 present data that appear to contradict the manuscript's emphasis on CpG as a uniquely targeted and selectively depleted dinucleotide in mammalian-adapted IAV. Several other dinucleotides, including UpC and GpG, exhibit similar variability across IAV infecting different hosts as CpG, raising questions about whether CpG content is uniquely constrained. The manuscript would benefit from additional discussion on broader dinucleotide usage trends and whether CpG-specific selection pressure is unique.

Reviewer #3:

This study by Becker et al examines the roles of ZAP and KHNYN in restricting viruses with RNA genomes high in CpG content. They show the mammalian versions of these proteins act independently as restriction factors against an engineered version of the PR8 strain of IAV that has elevated CpG content in one of the gene segments. They do not observe this effect with avian origin homologs. Based on the well documented observation that avian origin viruses tend to have higher CpG content and viruses that circulate in humans appear to be under selection to decrease CpG content, the authors argue that the high CpG content-dependence of ZAP/KHNYN-mediated restriction of PR8 reveals a general restriction factor that may act to protect mammals from avian origin viruses. They go on to show that the avian retrovirus ROSV is also sensitive to restriction by mammalian ZAP and KHNYN. Finally, they also clearly show that the restriction activities of ZAP and KHNYN are independent of each other.

The significance of the study comes from the in-depth molecular analysis of ZAP and KHNYN function and specificity and the clarification of their independent functions. The experiments are well designed and controlled, and the data generally support the conclusions of the paper. My only concern is that the differences in restriction between lab adapted PR8 and PR8cg are used to make fairly sweeping, general conclusions about species barriers. The language should be toned down at points and the limitations section fleshed out to reflect this.

Minor points:

Text on fig 2A hard to read

Fig 3C: any explanation for decreased replication in ZAP2 cells?

Line 356: PR8 is not a human IAV

---

## [Decision Letter · Decision Letter 2]

1 Oct 2025

Dear Jordan,

Thank you for your patience while we considered your revised manuscript "Mammalian ZAP and KHNYN can independently restrict CpG-enriched avian viruses" for publication as a Research Article at PLOS Biology. This revised version of your manuscript has been evaluated by the PLOS Biology editors, the Academic Editor and the original reviewers.

Based on the reviews, we are likely to accept this manuscript for publication, provided you satisfactorily address the remaining editorial requests. Please also make sure to address the following data and other policy-related requests.

a) We routinely suggest changes to titles to ensure maximum accessibility for a broad, non-specialist readership, and to ensure they reflect the contents of the paper. In this case, we would suggest a minor edit to the title, as follows. Please ensure you change both the manuscript file and the online submission system, as they need to match for final acceptance:

"Mammalian antiviral proteins ZAP and KHNYN can independently restrict CpG-enriched avian viruses"

b) Please add the link of the funding agencies in the Financial Disclosure statement in the manuscript details.

Please supply the numerical values either in the a supplementary file or as a permanent DOI’d deposition for the following figures:

Figure 1ACDFGH, 2A-D, 3CDE, 4ACDEFG, S1A-D, S2A-D, S3EFG, S4C-I, S7BC, S8A-E

d) Please cite the location of the data clearly in all relevant main and supplementary Figure legends, e.g. “The data underlying this Figure can be found in S1 Data” or “The data underlying this Figure can be found in https://doi.org/10.5281/zenodo.XXXXX”

e) Please ensure that you are using best practice for statistical reporting and data presentation. These are our guidelines https://journals.plos.org/plosbiology/s/best-practices-in-research-reporting#loc-statistical-reporting and a useful resource on data presentation https://journals.plos.org/plosbiology/article?id=10.1371/journal.pbio.1002128

-- If you are reporting experiments where n ≤ 5, please plot each individual data point.

f) We require the original, uncropped and minimally adjusted images supporting all blot and gel results reported in the Figures 2E-H, 3B, 4CDE, S3CD, S8A-E

-- We will require these files before a manuscript can be accepted so please prepare and upload them now. Please carefully read our guidelines for how to prepare and upload this data: https://journals.plos.org/plosbiology/s/figures#loc-blot-and-gel-reporting-requirements

We expect to receive your revised manuscript within two weeks.

*Published Peer Review History*

*Press*

Sincerely,

Melissa

Melissa Vazquez Hernandez, Ph.D.

Associate Editor

PLOS Biology

REVIEWERS' COMMENTS

Reviewer #1: The authors have addressed adequately the previous concerns

Reviewer #2: The authors have thoroughly and thoughtfully addressed all my prior concerns.

Reviewer #3: I feel the concerns raised by all three reviewers have been adequately addressed

---

## [Editor Report · Decision Letter 3]

16 Oct 2025

Dear Jordan,

Thank you for the submission of your revised Research Article "Mammalian antiviral proteins ZAP and KHNYN can independently restrict CpG-enriched avian viruses" for publication in PLOS Biology. I have taken over assessing your final version, in the absence of my colleague Melissa from the office, in order to prevent unnecessary loss of time. On behalf of my colleagues and the Academic Editor, Michaela Gack, I am pleased to say that we can in principle accept your manuscript for publication, provided you address any remaining formatting and reporting issues. These will be detailed in an email you should receive within 2-3 business days from our colleagues in the journal operations team; no action is required from you until then. Please note that we will not be able to formally accept your manuscript and schedule it for publication until you have completed any requested changes.

PRESS

Sincerely, 

Nonia

Nonia Pariente, PhD

Editor in Chief

PLOS Biology

on behalf of

Melissa Vazquez Hernandez, Ph.D.,

Associate Editor

PLOS Biology
